# Rab11b-mediated integrin recycling promotes brain metastatic adaptation and outgrowth

Erin N. Howe [1,2✉], Miranda D. Burnette[2,3,8], Melanie E. Justice [2,4], Patricia M. Schnepp [1,2,9],
Victoria Hedrick[5], James W. Clancy[1], Ian H. Guldner[1,2], Alicia T. Lamere[2,6,10], Jun Li[2,6], Uma K. Aryal [5],
Crislyn D'Souza-Schorey[1,2], Jeremiah J. Zartman [2,3] & Siyuan Zhang [1,2,7✉]

Breast cancer brain metastases (BCBM) have a 5-20 year latency and account for 30% of mortality; however, mechanisms governing adaptation to the brain microenvironment remain poorly defined. We combine time-course RNA-sequencing of BCBM development with a *Drosophila melanogaster* genetic screen, and identify Rab11b as a functional mediator of metastatic adaptation. Proteomic analysis reveals that Rab11b controls the cell surface proteome, recycling proteins required for successful interaction with the microenvironment, including integrin β1. Rab11b-mediated control of integrin β1 surface expression allows efficient engagement with the brain ECM, activating mechanotransduction signaling to promote survival. Lipophilic statins prevent membrane association and activity of Rab11b, and we provide proof-of principle that these drugs prevent breast cancer adaptation to the brain microenvironment. Our results identify Rab11b-mediated recycling of integrin β1 as regulating BCBM, and suggest that the recycleome, recycling-based control of the cell surface proteome, is a previously unknown driver of metastatic adaptation and outgrowth.

[1] Department of Biological Sciences, University of Notre Dame, Notre Dame, IN, USA. [2] Mike and Josie Harper Cancer Research Institute, University of Notre Dame, 1234 N. Notre Dame Avenue, South Bend, IN, USA. [3] Department of Chemical and Biomolecular Engineering, University of Notre Dame, Notre Dame, IN, USA. [4] Department of Chemistry and Biochemistry, University of Notre Dame, Notre Dame, IN, USA. [5] Purdue Proteomics Facility, Bindley Bioscience Center, Discovery Park, Purdue University, West Lafayette, IN, USA. [6] Department of Applied and Computational Mathematics and Statistics, University of Notre Dame, Notre Dame, IN, USA. [7] Indiana University Melvin and Bren Simon Cancer Center, Indianapolis, IN, USA. [8] Present address: Organogenesis, Birmingham, AL, USA. [9] Present address: Department of Urology, University of Michigan Medical School, Ann Arbor, MI, USA. [10] Present address: Mathematics Department, Bryant University, Smithfield, RI, USA. ✉email: ehowe2@nd.edu; szhang8@nd.edu

Breast cancer brain metastases (BCBMs) are an increasingly urgent clinical problem, with patient survival measured in months. Systemic treatments, such as chemotherapies or targeted therapies, cannot effectively treat micrometastatic brain lesions or prevent brain metastatic relapse, largely due to their inability to penetrate the blood–brain barrier (BBB)[1]. With better control of systemic disease, many women who have stable primary disease, or respond to initial treatment, ultimately develop BCBM. Thus, while survival of primary breast cancer is improving, the incidence of BCBM is increasing[2].

Metastasis is an inefficient process, with disseminated tumor cells (DTCs) dying at every stage of the process. Prior to the formation of overt metastatic disease, DTCs persist in the brain for months or years, and effective engagement with the metastatic niche is essential for colonization and outgrowth[3,4]. While the classic "seed and soil" hypothesis highlights the importance of optimal DTCs arriving in a permissive metastatic microenvironment, recent studies into metastatic seeding, dormancy, and outgrowth have revealed dynamic co-evolutionary processes between DTCs and the metastatic niche[5]. Indeed, successful metastatic adaptation requires organ-specific interactions with the surrounding parenchymal cells and extracellular matrix (ECM)[6], and the diversity of these interactions[7–11], combined with the evolving heterogeneity of the DTCs themselves, renders the mechanistic dissection required for the development of treatment options challenging. Once DTCs have adapted to the metastatic microenvironment and begun proliferating, current treatments often fail[12]; thus, identifying common mechanisms underlying the ability of DTCs to adapt to the metastatic microenvironment is critically important.

The ability of a DTC to successfully engage the metastatic microenvironment is dictated by the composition of the cell surface, which governs ligation of adhesion complexes, binding of growth factors, and engagement with parenchymal cells. Although much work has focused on transcriptional changes during tumorigenesis and metastasis, control of the cell surface through vesicular trafficking is emerging as a mechanism regulating several hallmarks of cancer[13]. Indeed, vesicular trafficking, including endocytosis and endosomal recycling, is the primary mechanism regulating the composition and organization of the cell surface[14]. Trafficking controls the localization and function of a variety of surface proteins with known roles in cancer and metastasis, including E-cadherin, EGFR, and integrins[15–17]. Yet, the role of trafficking, and the central machinery by which DTCs control the surface proteome in response to the metastatic microenvironment, remain poorly defined.

In this study, we identify Rab11b-mediated endosomal recycling as a unique mechanism for cancer cell adaptation to a challenging brain metastatic microenvironment. We first identify differentially regulated genes by utilizing RNA-sequencing to identify temporal changes during BCBM development. We then screen those genes for a functional role in brain metastasis using a *Drosophila melanogaster* tumor model[18], leading to the identification of Rab11b, a mediator of endosomal recycling. The Rab11 family of small GTPases is critical for recycling a number of proteins, and has been implicated in several types of cancer[19–21]. Perhaps the least well-studied family member, Rab11b localizes to the endosomal recycling center (ERC)[22], and is predominantly expressed in non-epithelial tissues, including brain[23]. We find that breast cancer cells up-regulate Rab11b during early adaptation to the brain metastatic site, providing a mechanism for DTCs to recycle needed proteins during this critical step of the metastatic cascade, enabling survival and outgrowth. Mechanistically, Rab11b-mediated control of the cell-surface proteome, including recycling of integrin β1, enables successful interaction with the brain ECM and mechanotransduction-activated survival signaling. Our findings suggest recycling controls the composition of the cell-surface proteome, which is critically important for metastatic cell-microenvironmental interaction and eventual outgrowth.

## Results

**Identification of functional mediators of brain metastasis.** To dissect temporal changes during breast cancer brain metastatic outgrowth, we analyzed the transcriptomes of early (7 days post injection, dpi) and late-stage overt brain metastases (40 dpi) using RNA-sequencing (Fig. 1a). Histology confirms the presence of colonized tumor cells in 7 dpi samples (Fig. 1b, black arrows). tdTomato-positive brain metastases were dissected from fresh brain tissue and sequenced, with 40 dpi samples split into three groups based on size at the time of dissection (small, medium, large). To exclude brain tissue-derived reads (mouse origin), only sequencing reads that uniquely mapped to the human genome were kept for downstream analysis (Supplementary Fig. 1a, b). We found that the 40 dpi brain metastases clustered away from 7 dpi samples, regardless of the size at dissection (Supplementary Fig. 1c), suggesting that metastatic adaptation and acquisition of a proliferative phenotype directs transcriptional reprogramming. Due to their similarity, we grouped 40 dpi samples together irrespective of size, and identified 125 genes that were significantly differentially regulated during breast cancer adaptation to the brain metastatic site with a Fisher's combined, Bonferroni-corrected $p$-value $< 0.05$ (Fig. 1c, Supplemental Table 1). Of the 125 dysregulated genes, 108 genes were up-regulated in overt brain metastasis (40 dpi) (Fig. 1d, BrainMets Sig.Genes).

To identify genes that functionally drive brain metastasis progression, we employed a *Drosophila melanogaster* tumor model for in vivo screening of the BrainMets Sig.Genes (Fig. 1d)[18]. This model overexpresses oncogenic Ras$^{V12}$, an RNAi construct targeting the polarity gene discs large (Dlg), and green fluorescent protein (GFP) in the epithelial *Drosophila* imaginal eye disc. In this model, tumors develop in the eye disc and progressively invade into adjacent brain tissue[24]. We identified *Drosophila* orthologs for the 108 BrainMets Sig.Genes, and obtained 448 RNAi fly lines (Supplementary Fig. 2a). A simple genetic cross drives expression of the RNAi construct specifically in the tumorous eye tissue (Fig. 1d, full genotype in Supplementary Fig. 2b). The integrated intensity of the tumor GFP signal was measured, and the effect size for each RNAi line was calculated as compared to the negative control RNAi line (yw, green) and the positive control (shPTEN, purple) using the strictly standardized mean difference (SSMD) (Fig. 1e). We identified 39 *Drosophila* RNAi lines representing 29 human genes that strongly suppressed tumor growth (Fig. 1e, Strong Negative SSMD, Supplementary Fig. 2c, 962—Rab11), while an additional 53 RNAi lines presenting 32 human genes moderately suppressed tumor growth (Fig. 1e, Moderate Negative SSMD, Supplementary Fig. 2c, 561—PSMC6). Interestingly, for the BrainMets Sig.Genes screened, gene expression was not strongly correlated with suppression of tumor growth (Supplementary Fig. 2d), suggesting that gene expression alone does not fully predict the functional importance of a gene in tumorigenesis and metastasis.

To account for variation in RNAi construct efficacy, we averaged SSMD scores for all RNAi lines targeting each gene to identify the genes that consistently decreased tumor growth and metastasis (Fig. 1f and Supplementary Fig. 2e). Genes with the lowest average integrated intensity SSMD scores (Aii.SSMD) strongly inhibited tumor growth. We identified the top 20 *Drosophila* genes based on Aii.SSMD, and examined the expression of their human homologs in samples from MDA-231 human breast cancer xenograft primary tumors and brain

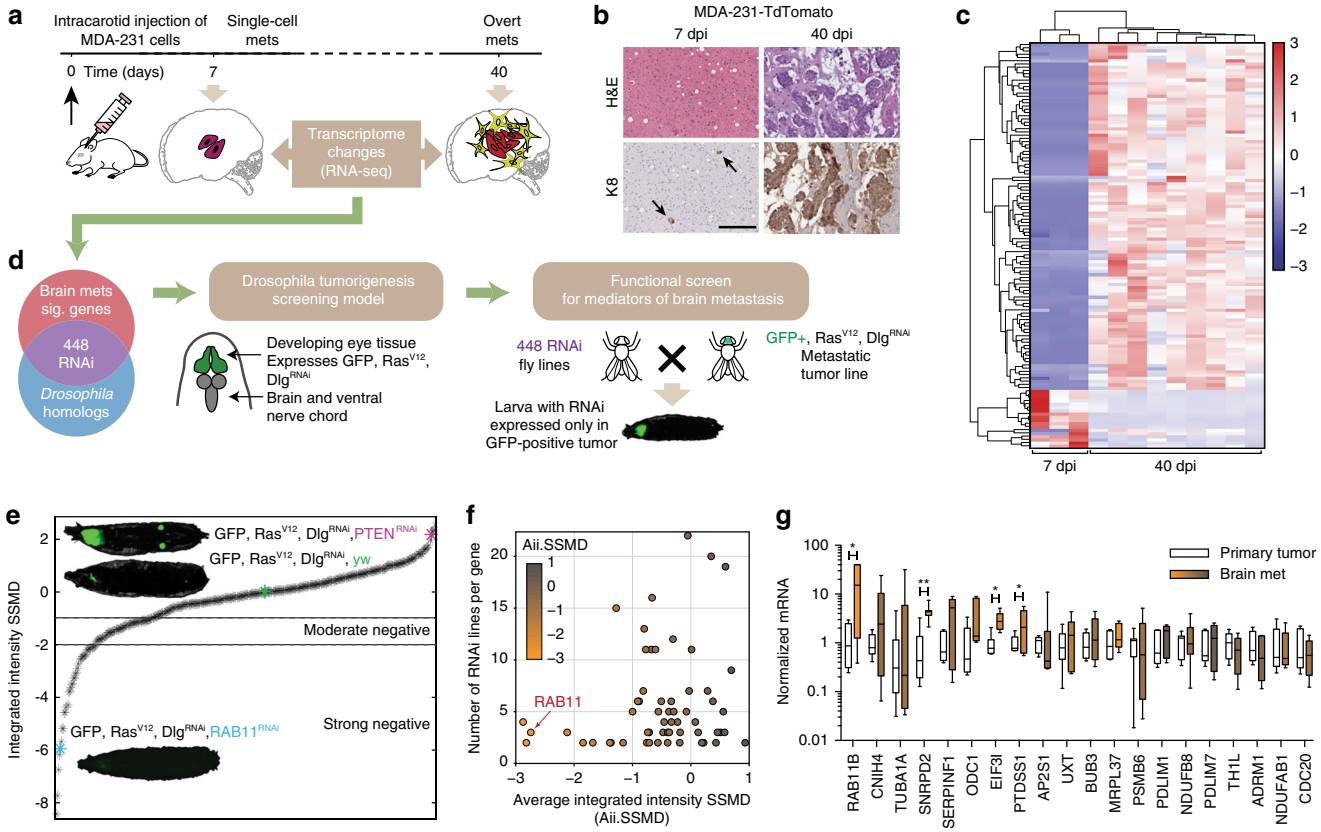

**Fig. 1 Identification of functional mediators of brain metastasis. a** Schematic of experimental design to generate temporal transcriptome of breast cancer brain metastases. **b** Representative images of brain mets at 7 or 40 dpi, as indicated. Tissues were H&E stained and cytokeratin 8 (K8) stained to show tumor tissue. Scale bar 200 μm. **c** Heatmap showing expression of 125 genes differentially expressed between 7 and 40 dpi. Fisher's combined test, Bonferroni-corrected $p$-value < 0.05. **d** Schematic of selection of *Drosophila* homologs, the genotype and phenotype of the *Drosophila* screening line, and the functional screening model used. **e** Data are presented as strictly standardized mean difference (SSMD) calculated with respect to the negative control (yw, no RNAi construct) shown in green, and the positive control (PTEN[RNAi]) shown in purple for a minimum of 15 larvae per cross. Hits are characterized as moderate or strong positives as indicated. Representative images for both controls, and a strong positive hit are shown. **f** For each RNAi line that yielded a strong, moderate or weak negative phenotype, the number of RNAi lines for that gene is plotted against the average integrated intensity SSMD. Datapoints are colored by average integrated intensity SSMD, with orange indicating a negative average SSMD, and gray indicating a positive average SSMD. **g** qRT-PCR for 20 genes with lowest average integrated intensity SSMD scores in MDA-231 primary mammary fat pad tumors (Primary Tumor, white), and brain metastases (Brain Met, orange to gray). Brain metastasis samples are colored to correspond to the gene's average integrated intensity SSMD, as in F. $n = 7$ animals/group. Boxes, first to third interquartile range, line, mean, whiskers, minimum and maximum values. Pairwise comparisons made as indicated using Student's $t$ test. $*p < 0.05$, $**p < 0.01$.

metastases. Among the top 3 *Drosophila* genes with the lowest Aii.SSMD (RAB11, SNRPD2, MRPL37), the human homolog of RAB11, RAB11B, was found to be increased 18-fold in brain metastasis tissue (Fig. 1g). We also identified SERPINF, a member of the SERPIN family, which was previously implicated in brain metastasis initiation[25], demonstrating the predictive power of our screening model for metastatic progression. In contrast with SERPINF1, which is moderately up-regulated in brain metastases, the highly inducible nature of RAB11B in brain metastases suggests a brain context-specific function of RAB11B (Fig. 1g).

**Rab11b is required for metastatic adaptation to the brain.** Rab11 is a family of small GTPases that regulate vesicular transport in the endosomal and exosomal recycling pathways, where they recycle a wide variety of proteins, including E-cadherin, epidermal growth factor receptor (EGFR), fibroblast growth factor receptor (FGFR), and multiple integrins[15,16,26]. Examination of a human tissue microarray containing 39 primary and 10 breast cancer brain metastasis samples confirms higher expression of Rab11 in brain metastases (Fig. 2a). While the *Drosophila* genome encodes a single Rab11, there are three

mammalian Rab11 family members: Rab11a, Rab25 (Rab11c), and Rab11b. Compared to cell line or primary tumors, the mRNA level of Rab11b, but not Rab11a or Rab25, is significantly up-regulated in brain metastases (Fig. 2b). Further, brain metastases derived from multiple breast cancer cell lines showed more than 15 fold increase of Rab11b expression compared with cultured cells (Fig. 2c, d), with no change in Rab11a or Rab25 levels (Supplementary Fig. 3a), suggesting the brain metastatic microenvironment specifically influences Rab11b expression. To determine if we can recapitulate the induction of Rab11b in vitro, we co-cultured breast cancer cells with primary murine glia, and found that only Rab11b is up-regulated (Fig. 2e, Supplementary Fig. 3b). Co-culture with murine caveolin 1[−/−] fibroblasts, a model of cancer-associated fibroblasts (CAFs)[27], showed no increase in Rab11b (Fig. 2e), confirming that the brain metastatic microenvironment uniquely induces expression of the Rab11b isoform.

To explore the temporal regulation of Rab11b during tumorigenesis or metastasis, we allowed MDA-231 cells to form primary tumors, lung metastases or brain metastases. We found that Rab11b is up-regulated in all three tissue microenvironments

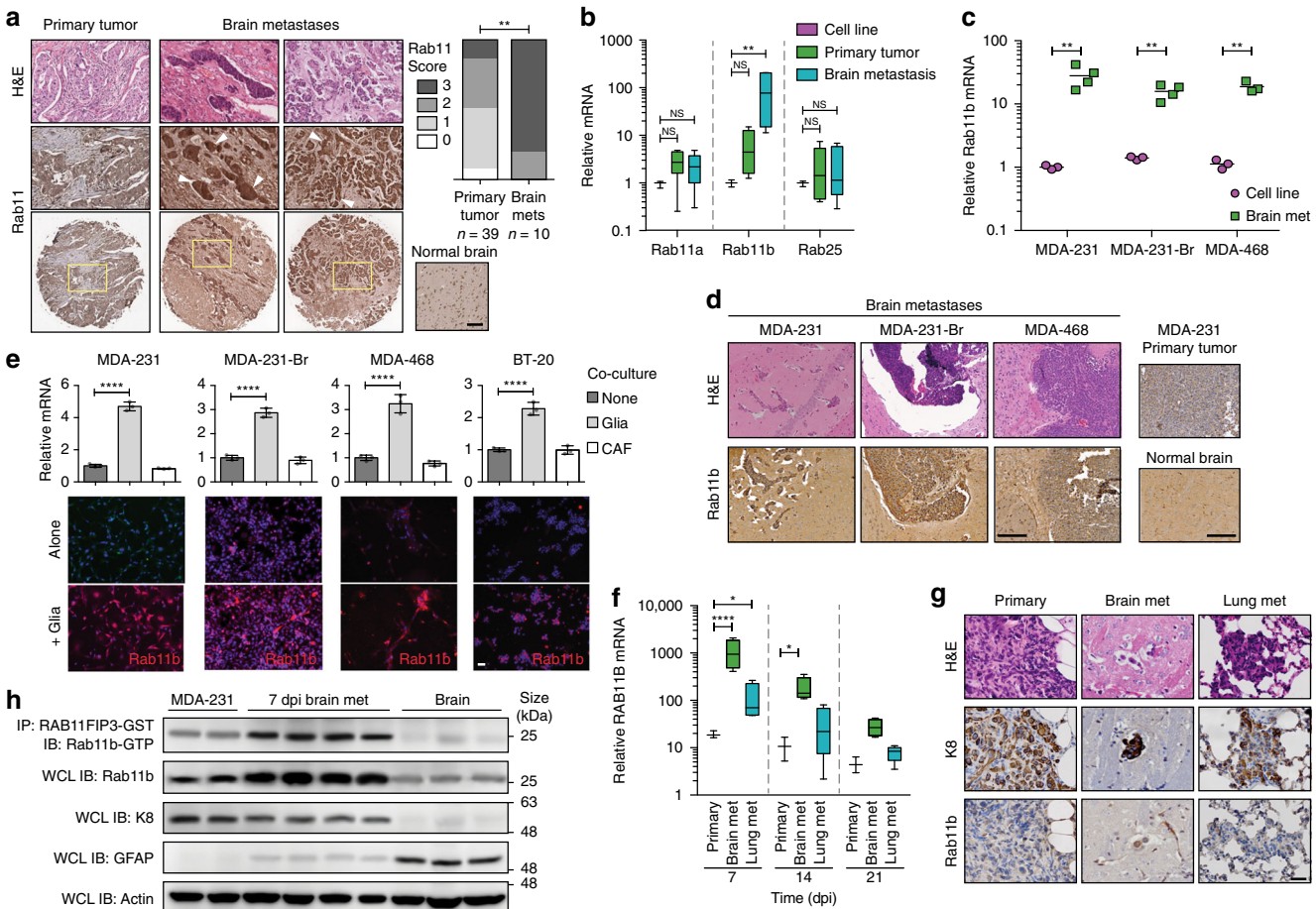

**Fig. 2 Rab11b is up-regulated during metastatic adaptation to the brain microenvironment. a** Representative images of H&E (top) or Rab11 immunohistochemical staining of human primary breast cancer or breast cancer brain metastases (arrowheads). Left, Rab11 IHC scoring. Analysis of contingency, Fisher's exact test. Scale bar 100 μm. **b** qPCR for Rab11 isoforms in MDA-231 cells grown in culture, primary tumors (21 dpi), or brain metastases (21 dpi). Values for each isoform normalized to cells in culture. $n = 3$ independent cell samples, 7 animals/group. Boxes, first to third interquartile range, line, mean, whiskers, minimum and maximum values. ANOVA, Dunnett's multiple comparison. **c** qPCR for Rab11b in cells in culture versus brain metastases. Values are normalized to MDA-231 cells in culture. $n = 3$ independent cell samples, 4 animals/group Line, mean. Student's $t$ test. **d** Representative H&E and Rab11b immunohistochemical staining of brain metastases, and MDA-231 primary tumor or murine brain. Scale bar 100 μm. **e** Top, qPCR for Rab11b in cell lines co-cultured with primary murine glia for 2 days. All values normalized to single culture. Bars, mean ± s.d. ANOVA, Dunnett's multiple comparison. Bottom, immunostaining for Rab11b (green, red) and nuclei (DAPI, blue) in cell lines cultured alone or co-cultured with primary murine glia for 5 days. $n = 3$. Scale bar 50 μm. **f** qPCR for Rab11b in MDA-231 primary tumors or metastases as indicated, collected at time points indicated. All values are normalized to MDA-231 cells in culture. $n = 3$. Boxes, first to third interquartile range, line, mean, whiskers, minimum and maximum values. Two-way ANOVA with Tukey's multiple comparison test. **g** Representative H&E, cytokeratin 8 (K8), and Rab11b immunohistochemical staining of MDA-231 primary tumors, brain or lung metastases, at 7 dpi. Scale bar 20 μm. **h** MDA-231 cells were intracranially injected and brain metastases dissected at 7 dpi using fluorescence signal as a guide. Metastases dissociated and brain parenchymal cells were removed using magnetic bead-based stromal cell depletion. Naïve brain (brain), or stroma depleted brain met (7 dpi brain met) samples were lysed and subjected to Rab11b activation assay, followed by immunoblotting. For all panels, *$p < 0.05$, **$p < 0.01$, ***$p < 0.001$, ****$p < 0.0001$.

early in tumor or metastasis formation; however, Rab11b is significantly more strongly up-regulated during early brain metastasis formation (Figs. 2f, 7 dpi). Rab11b immunohistochemistry confirms high expression of Rab11b protein specifically during early brain metastasis formation (Fig. 2g). To determine whether the induction of Rab11b expression leads to increased activation of Rab11b, we incubated naïve brain or dissected MDA-231 brain metastasis lysates with RAB11FIP3-GST, which specifically binds Rab11b-GTP[28]. We found a 76% increase in expression of total Rab11b, confirming our IHC results, as well as a 23% increase in the proportion of active Rab11b-GTP (Fig. 2h, Supplementary Fig. 3c, d). Taken together, these results suggest that breast cancer cells up-regulate Rab11b shortly after arriving in the brain metastatic microenvironment, and this up-regulation translates to increased Rab11b functional activity.

To determine whether Rab11b plays a causal role in breast cancer brain metastasis formation, we used two shRNA constructs targeting Rab11b and confirmed that they decrease Rab11b mRNA and protein (Fig. 3a, b, Supplementary Fig. 4a, b), as well as active Rab11b (Fig. 3c). To confirm that shRab11b constructs are able to block glial-mediated up-regulation of Rab11b, cells were co-cultured with primary murine glia, and glia were removed using magnetic-activated cell sorting prior to analysis. Although glia are able to induce up-regulation of Rab11b, the effect is severely blunted in cells expressing shRab11b constructs (Fig. 3d, e, Supplementary Fig. 4d, e). Given the high degree of homology between Rab11 family members, we sought to determine the specificity of the shRab11b constructs. We show that shRab11b-84 is specific to Rab11b (Supplementary Fig. 4c), while shRab11b-86 appears to moderately target Rab25 in MDA-

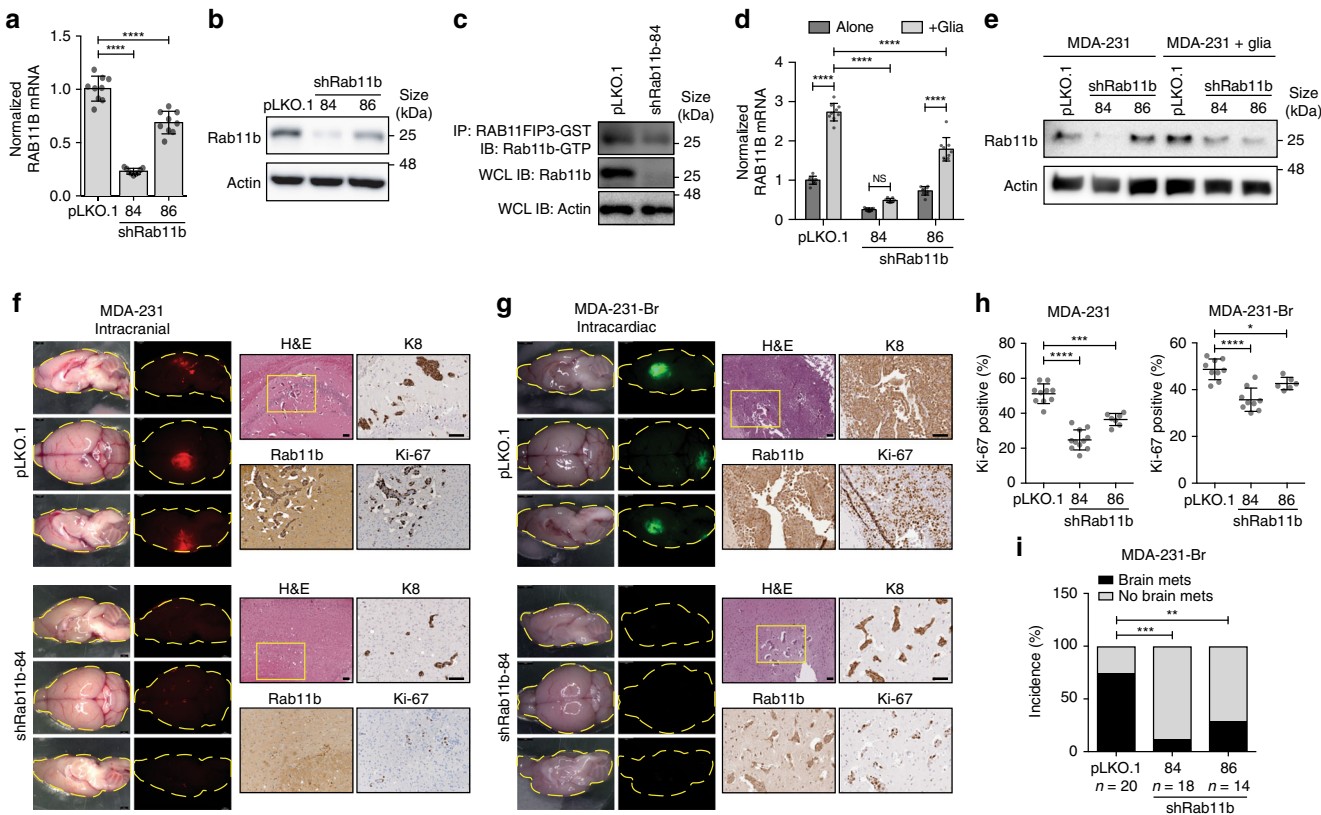

**Fig. 3 Rab11b is required for breast cancer brain metastasis. a** Normalized mean RAB11B expression in MDA-231 cells, relative to pLKO.1 empty vector. Three independent experiments. Bars, mean ± s.d. ANOVA, Dunnett's multiple comparison. **b** Rab11b immunoblots for MDA-231 cells expressing indicated constructs. **c** Rab11b-GTP immunoblot for MDA-231 cells. **d** Normalized RAB11B expression in MDA-231 cells cultured alone or with primary murine glia for three days, relative to pLKO.1 alone. n = 3 independent experiments. Bars, mean ± s.d. Two-way ANOVA, Tukey's multiple comparison. **e** Rab11b immunoblots for MDA-231 cells cultured alone or with primary murine glia for five days followed by removal of glial cells. **f** Representative H&E and IHC images for mice intracranially injected with MDA-231-tdTomato control or shRab11b cells. Scale bar 100 μm. **g** Representative H&E and IHC images for mice intracardially injected with MDA-231-Br-EGFP control or shRab11b cells. Scale bar 100 μm. **h** Quantitation of Ki-67 staining. n = 2 independent experiments. Bars, mean ± s.d. ANOVA, Dunnett's multiple comparison. **i** Incidence of MDA-231-Br brain metastasis determined by visible GFP signal at 28 dpi. Analysis of contingency, Fisher's exact test. For all panels, *p < 0.05, **p < 0.01, ***p < 0.001, ****p < 0.0001.

231 cells only (Supplementary Fig. 4c). Depletion of Rab11b dramatically decreases brain metastasis outgrowth in both intracranial (Fig. 3f, Supplementary Fig. 4f), and intracardiac (Fig. 3g, Supplementary Fig. 4g) injection models, with a significant reduction in proliferative ability (Fig. 3h). For intracardiac delivery of MDA-231-Br cells, we found that both shRab11b constructs also significantly decreased the incidence of brain metastasis formation (Fig. 3i). Taken together, these data demonstrate that Rab11b enhances brain metastasis formation following lodging in the brain parenchyma through specific interactions with the brain microenvironment, and loss of Rab11b is sufficient to decrease breast cancer brain metastasis.

**Recycling controls the surface proteome.** Given the role of Rab11b in recycling cell-surface proteins, we examined Rab11b-mediated recycling of the transferrin receptor (TfR), a canonical Rab11 cargo protein[29]. We examined the rate at which TfR was recycled by assaying the amount of fluorescently labeled transferrin retained in cells following 1 h loading with fluorescent transferrin. Compared with control cells, loss of Rab11b led to slower TfR recycling (Fig. 4a, Supplementary Fig. 6a), although TfR was fully recycled in all cell lines examined, consistent with the requirement for TfR recycling in cellular homeostasis. Internalization of TfR was not affected (Supplementary Fig. 6b), consistent with the role of Rab11b in regulating transport of proteins from the ERC to the cell surface[22]. These results confirm

that shRab11b functionally perturbs endosomal recycling. Interestingly, there is not a detectable increase of TfR recycling when cells are cultured with either primary glia or CAF cells (Fig. 4b, Supplementary Fig. 6c), despite the strong glial-mediated induction of Rab11b (Fig. 2e). This suggests that brain-mediated up-regulation of Rab11b might regulate a specific subset of cell-surface proteins, rather than globally altering recycling of all cargo proteins.

To examine Rab11b-regulated retention and recycling of surface proteins, we biotinylated all cell-surface proteins (Fig. 4c). Rab11b associates with internalized surface proteins, with shRab11b cells showing decreased association (Fig. 4d). There was a small decrease in total cellular retention of biotinylated proteins in shRab11b cells (Fig. 4e, left), but a significant decrease in surface expression of biotinylated proteins (Fig. 4e, right), suggesting an inability of shRab11b cells to return a fraction of internalized surface proteins to the surface. To identify the subset of surface proteins that are dependent on Rab11b for recycling, we performed mass spectrometry analysis of the cell-surface proteomes of control and shRab11b cells (Fig. 4f). We observed a strong correlation between biological replicates (Fig. 4g). Although equal amounts of surface protein were loaded (Supplementary Fig. 6a), the number of proteins identified in shRab11b cells was lower, 234 ± 32 proteins for shRab11b cells versus 626 ± 102 for control cells (Supplementary Fig. 7b), suggesting that there is decreased diversity in the cell-surface

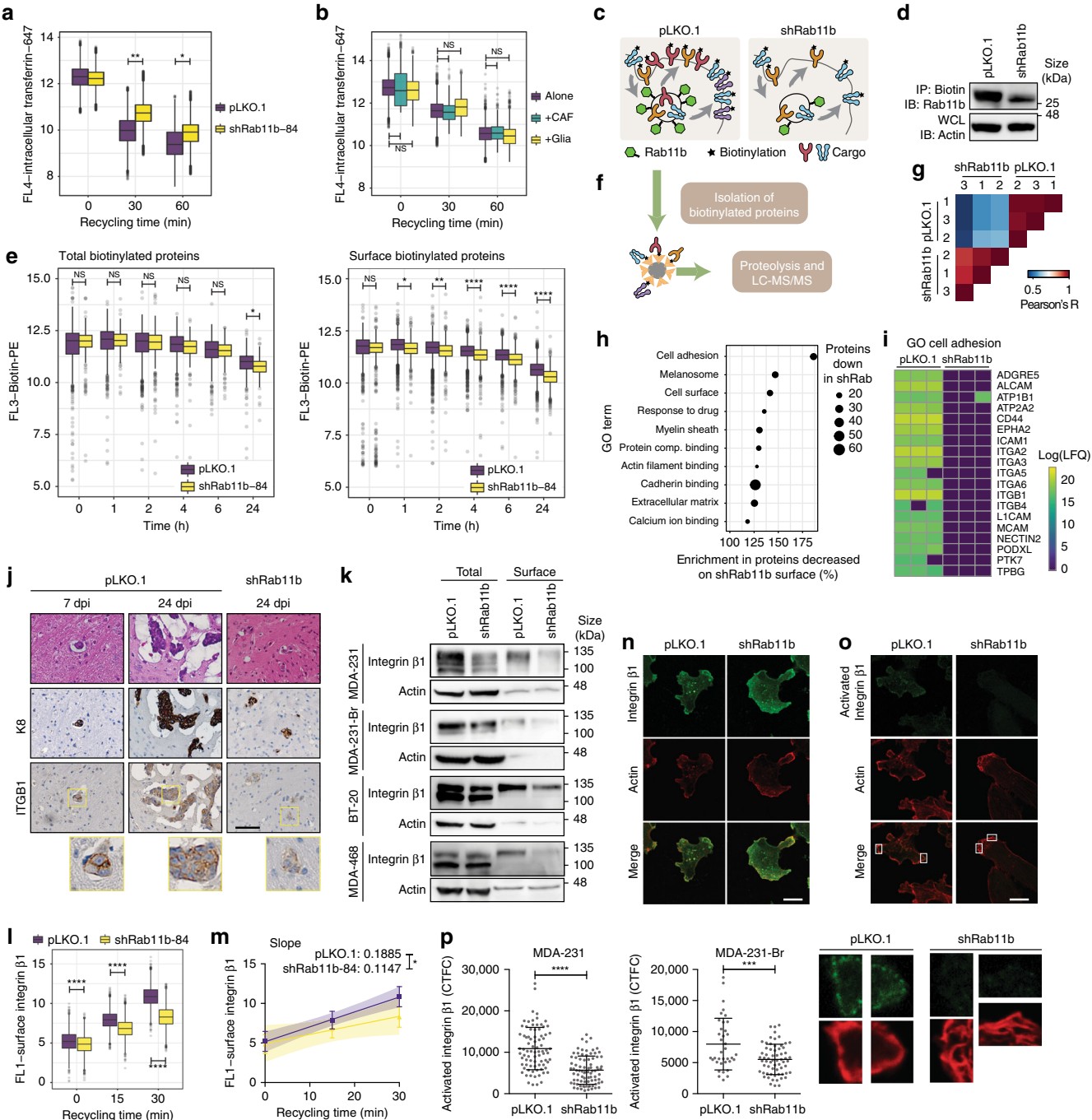

**Fig. 4 Rab11b recycling alters the surface proteome and controls integrin β1 localization and activation. a** Transferrin receptor recycling in MDA-231 cells. $n = 3$ independent experiments. Two-way ANOVA, Sidak's multiple comparison. **b** Transferrin receptor recycling in MDA-231 cells co-cultured for two days. Cancer cells were selected on expression of CD-44. $n = 3$ independent experiments. Two-way ANOVA, Sidak's multiple comparison. **c** Schematic of surface biotinylation. **d** Association of Rab11b with biotinylated surface proteins determined by immunoprecipitation and immunoblotting. **e** Retention of total or surface biotin following surface biotinylation with biotin-PE. $n = 2$ independent experiments. Two-way ANOVA, Sidak's multiple comparison. **f** Schematic of biotinylated surface protein isolation and proteomics. **g** Correlation matrix of all measured samples based on Pearson's correlation values. **h** Cleveland plot of top GO terms enriched in proteins that were decreased on the surface of shRab11b cells. **i** Heatmap of proteins annotated with GO term cell adhesion, containing at least one predicted transmembrane domain. **j** Representative images of H&E and IHC for mice intracranially injected with MDA-231-tdTomato cells. IHC for cytokeratin 8 (K8), Rab11b and integrin β1 (ITGB1). Scale bar 50 μm. **k** Immunoblotting of total and surface lysates. **l** Surface integrin β1 recycling in MDA-231 cells. $n = 2$ independent experiments. Two-way ANOVA, Sidak's multiple comparison. **m** Linear regression of data in (**p**). Points, mean ± s.d., color shading, 95% confidence interval. Analysis of covariance. **n–p** MDA-231 cells stained for integrin β1 and actin (phalloidin, to delineate cell boundaries) (**n**), or active integrin β1 and actin (**o**). **p** Corrected total cellular active integrin β1 fluorescence (CTCF) was determined for individual cells. $n = 3$ independent experiments. Bars, mean ± s.d. Two-sided t-test. Scale bar 15 μm. For panels (**a**, **b**, **e**, and **l**) Boxes, first to third interquartile range, line, mean, points, outliers. For all panels, *$p < 0.05$, **$p < 0.01$, ***$p < 0.001$, ****$p < 0.0001$.

proteome of cells without Rab11b. Bioinformatic analysis of known surface localized proteins yielded 226 proteins in pLKO.1, and 64 proteins in shRab11b cells (Supplementary Fig. 7c). To identify protein functional groups, the incidence of repeated GO terms was determined for proteins decreased or lost from the surface of shRab11b cells, which revealed that proteins involved in cell adhesion are dramatically decreased in shRab11b cells (Fig. 4h). Further analysis of proteins involved in cell adhesion, combined with the identification of proteins with transmembrane domains using the prediction algorithm TMHMM[30], revealed that integrins in particular are dramatically decreased on the surface of shRab11b cells (Fig. 4i).

**Rab11b controls integrin β1 localization and activation.** Integrins are heterodimeric proteins that mediate cell attachment to the ECM[31]. Of the integrins identified in our cell-surface proteome analysis, integrin β1 (ITGB1) is the most versatile[32], recognizing a range of ECM proteins, and forming heterodimers with the majority of α integrins, including all of the α integrins we found to be decreased on the surface of shRab11b cells (Fig. 4i). Examination of integrin β1 revealed strong expression in brain metastases, with reduced expression in shRab11b brain metastases (Fig. 4j). Examination of breast cancer cells in culture reveals that knocking down Rab11b slightly decreases overall expression of integrin β1, but leads to dramatic loss of cell-surface integrin β1 (Fig. 4k), suggesting a loss of localization. To determine if loss of Rab11b alters integrin β1 recycling, surface integrin β1 was antibody labeled and cells were suspended for 1 h to down-regulate adhesion signaling and allow internalization of integrin β1. Analysis of labeled integrin β1 recycled to the surface reveals that control cells rapidly recycle integrin β1, while shRab11b cells exhibit lower surface levels (Fig. 4l) and the rate of integrin β1 recycling is significantly decreased (Fig. 4m). Next, we examined total and active integrin β1 and found that while total integrin β1 expression was not altered in shRab11b cells (Fig. 4n), there was an almost complete loss of activated integrin β1 (Fig. 4o). Quantification of active integrin β1 staining confirms a dramatic decrease in shRab11b cells (Fig. 4p), and siRab11b cells (Supplementary Fig. 5a–c). Taken together, these data suggest that Rab11b-mediated recycling controls the cell-surface proteome, maintaining integrin β1 surface localization, where it can be activated by the ECM to facilitate tumor cell survival.

**Integrin β1 facilitates tumor cell survival in the brain.** Successful ECM engagement is required for the survival of cancer cells, particularly metastatic cancer cells as they adapt to a new metastatic microenvironment[33]. The loss of surface integrin β1 suggests that shRab11b cells would exhibit decreased adhesion to integrin β1 ligands. Indeed, we found that loss of Rab11b led to delayed ECM engagement (Fig. 5a), although they were ultimately able to adhere (Supplementary Fig. 8a). To determine the extent of reliance on integrin β1, we plated cells on poly-L-lysine (pLL), which does not induce integrin activation, or Type I collagen, an integrin β1 ligand. Control and shRab11b cells remain rounded on pLL, although control cells are better able to attach and spread (Fig. 5b). On collagen, control cells successfully spread and assume their characteristic mesenchymal-like morphology, while shRab11b cells showed an evident defect in spreading with reduced cell area and length (Fig. 5b). The loss of cell spreading was confirmed in siRab11b cells (Supplementary Fig. 5d, e). Taken together, this suggests that loss of integrin β1 from the surface of shRab11b severely reduces cell spreading, and further that control cells possess a weak, non-integrin β1-mediated mechanism for spreading, which is likewise lost in shRab11b cells. To examine signaling downstream of integrin-mediated

attachment, we suspended cells for 1 h to force down-regulation of adhesion-mediated signaling, and found that shRab11b induced a severe defect in post-attachment activation of focal adhesion kinase (FAK) (Fig. 5c). Control cells exhibit sustained phosphorylation at residues 397 and 925 for up to 6 h (Fig. 5c, Supplementary Fig. 8b), demonstrating auto-phosphorylation and activation of FAK. shRab11b cells exhibit decreased phosphorylation at all of the sites, including residue 925, which is involved in cell migration and protrusions[34], suggesting that shRab11b cells fail to spread (Fig. 5b) due to decreased focal adhesion formation. Although this reduction in adhesion-mediated signaling does not ultimately change the proliferation of shRab11b cells in culture (Supplementary Fig. 8c), loss of Rab11b dramatically decreases both colony size and number when cells are grown in soft agar (Fig. 5d). These data suggest that the adhesion defects mediated by loss of Rab11b are masked when cells are grown in idealized cell culture conditions, likely due to the eventual ability of cells to adhere. However, Rab11b driven defects become apparent when cells are exposed to micro-environmental stress, such as during growth in soft agar. We next examined the effect of shRab11b on FAK and Erk signaling during adhesion to an integrin β1 ligand. When control and shRab11b cells are plated on pLL, although both lines have low activation, shRab11b cells exhibit decreased activation of FAK and Erk (Fig. 5e), consistent with their decreased ability to spread. Across multiple cell lines, control cells dramatically increase activation of FAK and Erk when plated on Col I, while shRab11b cells exhibit decreased phosphorylation of FAK and Erk, suggesting that loss of surface integrin β1 induces a defect in ligation to Col I leading to decreased focal adhesion-mediated Erk signaling (Fig. 5e). To further investigate the role of Rab11b in mediating integrin β1 signaling, we treated control cells with an inhibitory integrin β1 antibody (P5D2), or shRab11b cells with an activating antibody (12G10). Inhibition of integrin β1 in control cells induces auto-phosphorylation of FAK[35], but decreases activation of Erk (Fig. 5f), while activation of integrin β1 in shRab11b cells increases FAK and Erk signaling (Fig. 5f). Furthermore, modulation of integrin β1 activation status also leads to suppression (P5D2), or induction (12G10) of growth in soft agar in control and shRab11b cells, respectively (Fig. 5g, Supplementary Fig. 8d). Together, these data suggest that Rab11b-mediated recycling of integrin β1 enhances the ability of breast cancer cells to successfully attach, activate attachment-mediated signaling, and ultimately survive in sub-optimal ECM conditions.

As cells metastasize, they are exposed to new ECM conditions, and their ability to successfully engage this foreign ECM is critical for metastatic success. To determine whether Rab11b directly impacts the ability of breast cancer cells to engage the brain ECM, we cultured tumor cells on decellularized murine brain ECM[36]. At 2 h the cells have begun to adhere, and the equal proliferative rates of control and shRab11b cells in culture are apparent; however, as cells require adhesion-mediated signaling in response to the brain ECM, shRab11b cells are unable to initiate (24 h) and sustain (48 h) cell proliferation (Fig. 5h). Immunostaining reveals considerably decreased active integrin β1 in shRab11b cells plated on brain ECM (Fig. 5i, Supplementary Fig. 8e). Phalloidin staining reveals that control cells are able to spread and form protrusions on the brain matrix, while shRab11b cells remain rounded up (Fig. 5j, left). Inhibition (P5D2) or activation (12G10) of integrin β1 prevents or induces spreading and protrusion formation in control and shRab11b cells, respectively (Fig. 5j, k). Further, inhibiting integrin β1 signaling prevents proliferation in control cells, while activation induces proliferation of shRab11b cell on brain matrix (Fig. 5l). Taken together, these data suggest that Rab11b is required for surface localization of integrin β1, which in turn initiates focal adhesion signaling, allowing

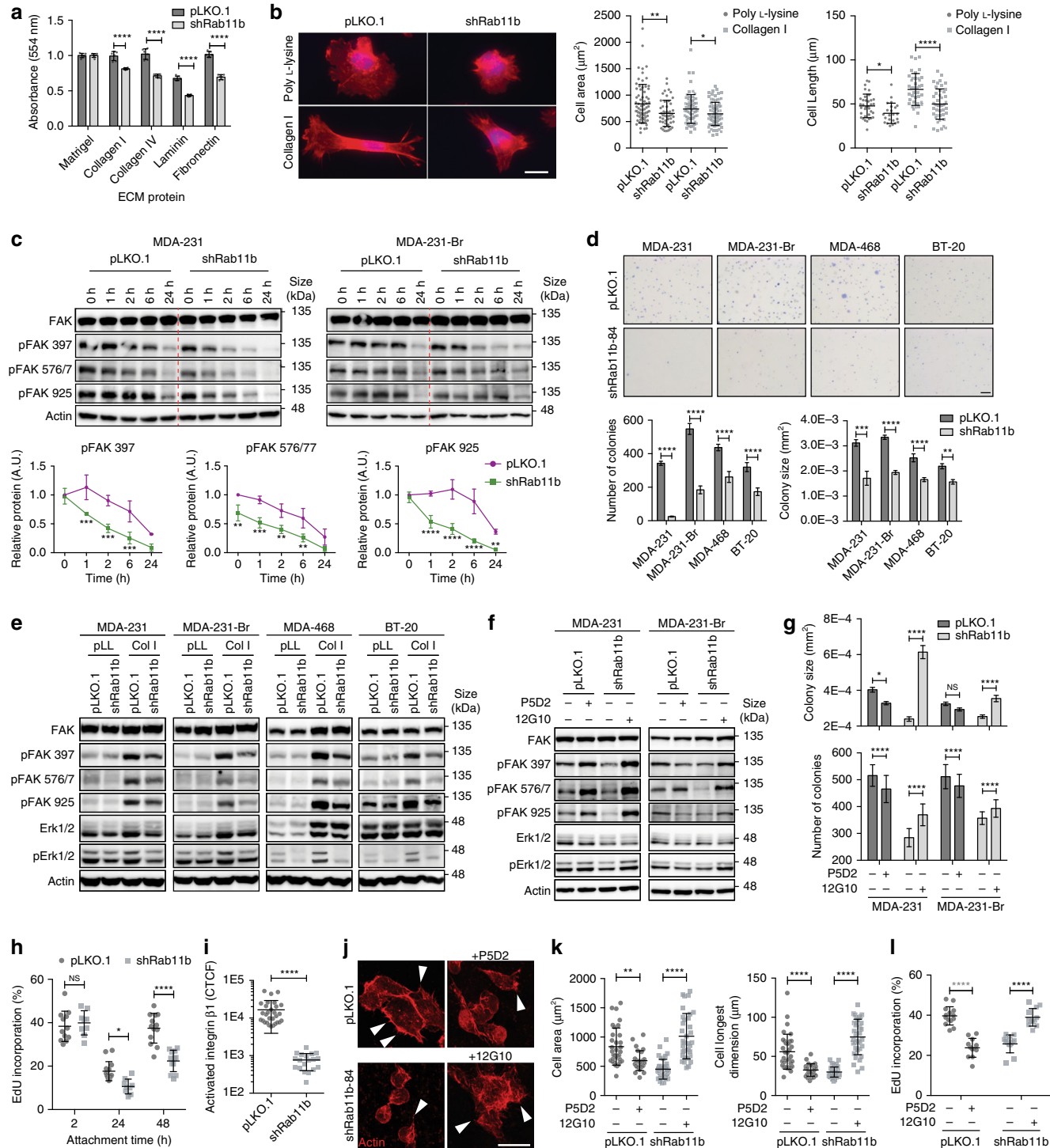

attachment to the brain ECM and ultimately survival in the brain metastatic microenvironment.

**Inhibition of Rab11b activity decreases brain metastasis.** Given the role of Rab11b in breast cancer brain metastasis, we sought to pharmacologically inhibit Rab11b. Historically, targeting small GTPases has not been easily accomplished or translated to the clinic[37]. However, a unique feature of Rab proteins is that they require geranylgeranylation for membrane localization, which is required for function[38]. Geranylgeranylpyrophosphate is generated by the mevalonate pathway (Fig. 6a)[39], which is inhibited by HMG-CoA reductase inhibitors such as statins[40]. Thus, we

postulate that through inhibition of the mevalonate pathway, statin treatment could also suppress Rab11b activation, which we have shown is an essential step for breast cancer adaptation to the brain metastatic microenvironment. Because brain metastases, particularly early brain metastases, are protected by the BBB[41], we chose two lipophilic statins, pitavastatin (Pit) and simvastatin (Sim), which are BBB permeable[42,43]. We found that both statins successfully inhibited tumor cell growth in soft agar (Fig. 6b), phenocopying the effects of Rab11b knockdown (Fig. 5d).

Pitavastatin and simvastatin have both been shown to inhibit breast cancer proliferation in vitro and in vivo by inhibiting RhoA, NF-κB, Arf6 or PI3K signaling[44–47]. To determine if statin

**Fig. 5 Rab11b recycling of integrin β1 is necessary for survival in the brain microenvironment. a** Cell attachment at 6 h, normalized to Matrigel control. $n = 3$ independent experiments. Bars, mean ± s.d. Two-way ANOVA, Sidak's multiple comparison. **b** Representative images of actin (phalloidin, red) and nuclei (DAPI, blue) for cells plated on poly-ʟ-lysine or Collagen I for 24 h. Scale bar 20 μm. Right, quantification of cell area and longest dimension. $n = 3$ independent experiments. Bars, mean ± s.d. ANOVA, Tukey's multiple comparison. **c** Cells suspended for 1 h at 37 °C, then plated to allow adhesion complex formation. Immunoblots showing signaling during adhesion. Bottom, quantification of pFAK in MDA-231, presented relative to FAK normalized to actin. $n = 3$. **d** Top, representative images of cells grown in soft agar for three weeks. Bottom, quantification of colony number and size. $n = 10$ fields per 3 independent experiments. Scale bar 1 mm. **e** Immunoblots showing signaling after 6 h adhesion to poly-ʟ-lysine (pLL) or Collagen I (Col I). **f** Immunoblots showing signaling after 1 h incubation with P5D2 or 12G10, followed by 6 h adhesion to Col I. **g** Quantification of colony number and size for cells treated with P5D2 or 12G10 and grown in soft agar for two weeks. Ten fields per condition. **h** Quantification of EdU incorporation for cells adhering to decellularized brain matrix. Ten fields per condition. **i** Corrected total cellular active integrin β1 fluorescence (CTCF). $n = 3$ independent experiments. Bars, mean ± s.d. Two-sided t-test. (**j–l**). **j** Representative images of actin (phalloidin, red) protrusions (arrowheads) for cells treated with P5D2 or 12G10 and allowed to adhere to decellularized brain matrix for 48 h. Scale bar 25 μm. **k** Actin staining used to quantify cell area and longest dimension. Bars, mean ± s.d. ANOVA, Tukey's multiple comparison. **l** Quantification of EdU incorporation. Ten fields per condition. Bars, mean ± s.d. ANOVA, Tukey's multiple comparison. For panels (**c**, **d**, **g**, **h**), bars, mean ± s.d. Two-way ANOVA, Sidak's multiple comparison. For all panels, *$p < 0.05$, **$p < 0.01$, ***$p < 0.001$, ****$p < 0.0001$.

treatment decreases tumorigenicity in part through inhibition of Rab protein activity, we performed a metabolite rescue experiment. Cells were treated with statins in addition to mevalonic acid (MVA), which restores all signaling downstream of HMG-CoA, including geranylgeranylation of Rab proteins. Restoration of the entirety of the mevalonate pathway with MVA rescues growth in soft agar as expected (Fig. 6c, Supplementary Fig. 9a). Importantly, addition of geranylgeranylpyrophosphate, a metabolite essential for geranylgeranylation of Rab proteins, significantly rescued growth of tumor cells in soft agar, increasing both colony size and number (Fig. 6c, Supplementary Fig. 9a). We next sought to determine if there was a synergistic effect between statin treatment and Rab11b knockdown. In soft agar assays, we found a small but insignificant decrease in colony size, with no decrease in colony number (Fig. 6d, Supplementary Fig. 9b). Taken together, this suggests statins mediate their effect on tumorigenicity in part through inhibition of Rab11b.

To determine if statins decrease Rab11b geranylgeranylation, we examined the membrane localization of Rab11b. Both pitavastatin and simvastatin moved Rab11b from the insoluble, membrane bound fraction to the soluble, non-membrane bound fraction in a dose dependent manner (Fig. 6e, Supplementary Fig. 9c), suggesting a loss of Rab11b membrane localization. Consistent with loss of membrane localization, statin treatment also decreased the amount of active Rab11b (Fig. 6f), and induced a dose dependent decrease in the rate of transferrin receptor recycling (Fig. 6g). Taken together, thse data suggest that both pitavastatin and simvastatin effectively inhibit geranylgeranylation of Rab11b, thereby preventing membrane localization, activation and function. We next sought to determine if statins would prevent integrin β1 recycling and adhesion-mediated signaling. Both pitavastatin and simvastatin decreased the rate at which internalized integrin β1 is recycled to the cell surface (Fig. 6h). When cells are plated on decellularized brain matrix, statin treatment prevents cell spreading and protrusion formation (Fig. 6i, j), decreases activation of integrin β1 (Fig. 6k), and inhibits proliferation of breast cancer cells on the brain ECM (Fig. 6l), suggesting a role for integrin β1 downstream of mevalonate pathway inhibition. To confirm the role of integrin β1, we treated cells with statins alone or in conjunction with an integrin β1 activating antibody (12G10). We found that activation of integrin β1 restores cell spreading and protrusion formation on decellularized brain matrix (Fig. 6m, n), as well as proliferation (Fig. 6o). Taken together, these data suggest that both pitavastatin and simvastatin successfully inhibit Rab11b, leading to decreased recycling and activation of integrin β1.

To confirm this effect in vivo, we used both intracardiac and intracranial models of breast cancer brain metastasis. Beginning two days post injection, animals given daily treatment with

human equivalent doses of pitavastatin or simvastatin showed a dramatic decrease in brain metastasis incidence (Fig. 7a, d), with a concurrent increase in survival (Fig. 7c). Brain metastases also exhibited decreased proliferation (Fig. 7b, e, Supplementary Fig. 9d). Given that statins inhibit membrane localization of Rab11b (Fig. 6e), statin treatment did not alter Rab11b expression as expected (Fig. 7g). However, statin treatment significantly decreases the expression of integrin β1 in brain metastases (Fig. 7f, g), suggesting that statin treatment inhibits Rab11b-mediated recycling of integrin β1 to suppress breast cancer brain metastasis. Taken together, thse data show that pitavastatin and simvastatin are able to inhibit Rab11b activity, leading to decreased recycling of integrin β1, and ultimately suppressing the ability of breast cancer cells to successfully engage the brain metastatic ECM.

## Discussion

Metastatic disease is an undisputedly urgent clinical problem, and patients with brain metastases protected by the BBB are in particular need of effective therapeutic options. We provide here two significant insights into breast cancer brain metastasis. First, whereas metastatic adaptation is often studied at the transcriptome level, our data show that endosomal recycling can dramatically alter the ability of DTCs to interact with their microenvironment through control of protein localization to the cell surface. Several recent studies combining proteomic and genomic analysis have demonstrated a low degree of concordance between mRNA and protein expression[34,48,49], highlighting the importance of protein-level regulation, such as recycling. Second, we demonstrate the pre-clinical efficacy of statin treatment for BCBM, and provide a mechanistic rationale for this efficacy, through inhibition of Rab11b geranylgeranylation.

By combining temporal transcriptional analysis of breast cancer brain metastasis formation with functional screening in a model organism, we identify genes that are not only transcriptionally dysregulated, but functionally driving tumorigenesis and metastasis. We show that breast cancer cells significantly up-regulate Rab11b in the brain metastatic site compared to the primary site. Previous studies have identified diverse mechanisms controlling breast cancer survival in the brain metastatic microenvironment, often relying on transcriptional profiling of cancer or microenvironmental cells[50]. Our finding that endosomal recycling exerts control over the cell-surface proteome suggests that it is an important, yet previously unconsidered, level of regulation coordinating metastatic adaptation.

Specifically, we show that loss of Rab11b decreases retention of proteins on the cell surface, with a dramatic loss of several integrins, including integrin β1. It is well known that localization governs protein function, particularly for adhesion proteins and

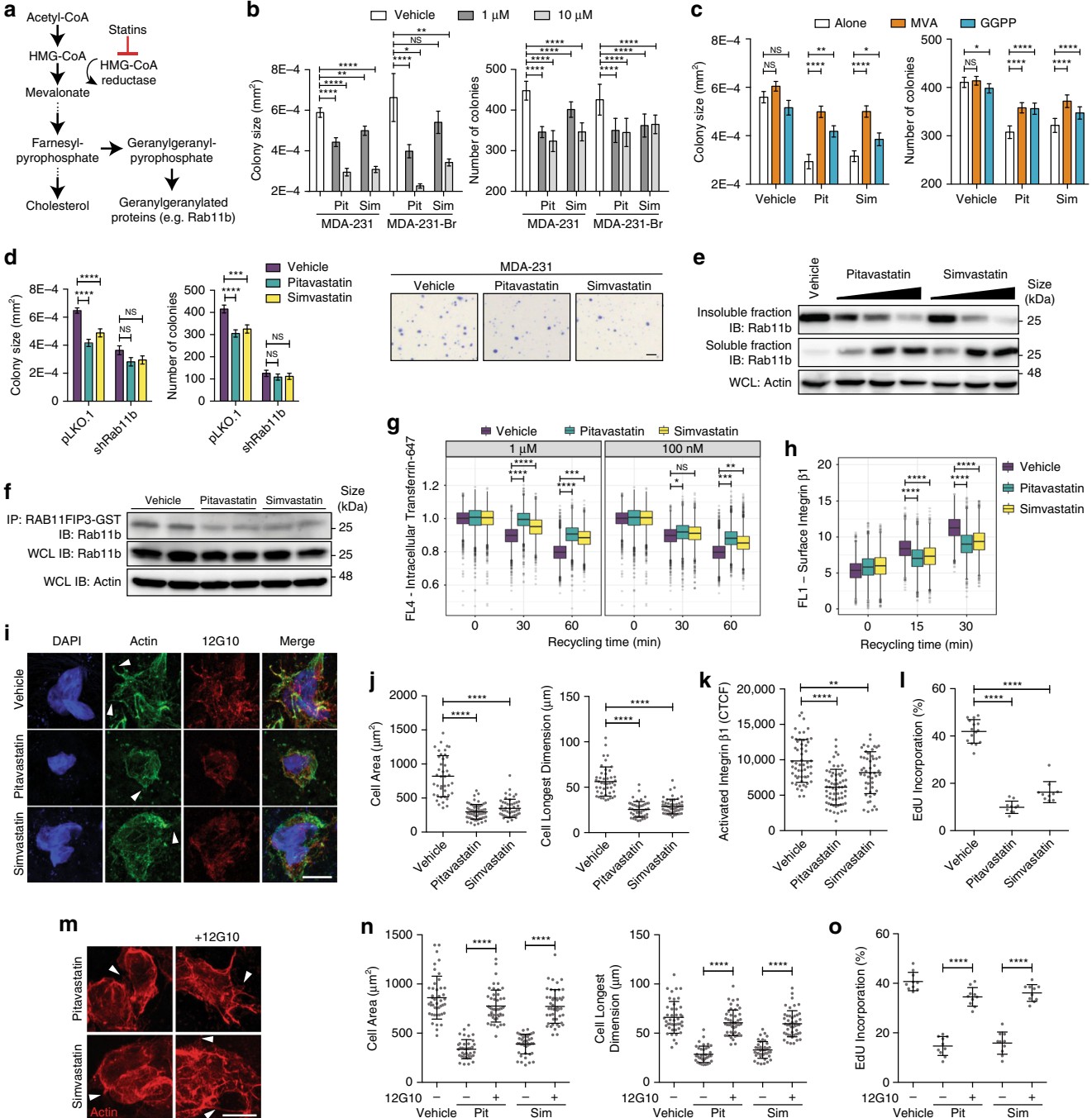

**Fig. 6 Statins decrease Rab11b localization and function. a** Schematic of the mevalonate pathway leading to Rab11b geranylgeranylation. **b** Quantification of cells grown in soft agar. Bottom, representative images. Scale bar 1 mm. **c** Quantification of colony size and number for MDA-231 cells grown in soft agar with 1 μM pitavastatin/simvastatin, with 100 μM mevalonic acid or 10 μM geranylgeranylpyrophosphate. **d** Quantification of MDA-231 cells grown in soft agar with vehicle or 1 μM pitavastatin or simvastatin. **e** MDA-231 cells grown with vehicle or 10 μM-100 nM pitavastatin/simvastatin for 24 h. Immunoblotting of soluble and insoluble fractions separated with Triton X-114. **f** Rab11b activation assay for MDA-231 cells treated with vehicle or 1 μM pitavastatin/simvastatin for 24 h, followed by immunoblotting. **g** Transferrin receptor recycling in MDA-231 cells treated with vehicle or 1 μM pitavastatin or simvastatin. **h** Surface integrin β1 recycling in MDA-231 cells treated with vehicle or 1 μM pitavastatin/simvastatin. (**i-l**) MDA-231 cells treated with 1 μM pitavastatin or simvastatin and adhered to decellularized murine brain matrix for 48 h. (I) Representative images of actin (phalloidin, green) protrusions (arrowheads), active integrin β1 (12G10, red), and nuclei (DAPI, blue). Scale bar 10 μm. **j** Quantification of cell area and longest dimension. **k** Corrected total cellular active integrin β1 fluorescence (CTCF). **l** Quantification of EdU. **m-o** MDA-231 cells treated with 1 μM pitavastatin or simvastatin and 12G10, and adhered to decellularized murine brain matrix for 48 h. **m** Representative images of actin (phalloidin, red) protrusions (arrowheads). Scale bar 10 μm. **n** Quantification of cell area and longest dimension. **o** Quantification of EdU. For panels (**b**, **c**), n = 10 fields per 3 independent experiments. Bars, mean ± s.d. Two-way ANOVA, Sidak's multiple comparison. For panels **g**, **h**, n = 2 independent experiments. Boxes, first to third interquartile range, line, mean, points, outliers. Two-way ANOVA, Sidak's multiple comparison. For panels (**j**, **k**, **n**), n = 3 independent experiments. Bars, mean ± s.d. ANOVA, Tukey's multiple comparison. For panels (**l**, **o**), n = 10 fields per 3 independent experiments. Bars, mean ± s.d. ANOVA, Tukey's multiple comparison. For all panels, *p < 0.05, **p < 0.01, ***p < 0.001, ****p < 0.0001.

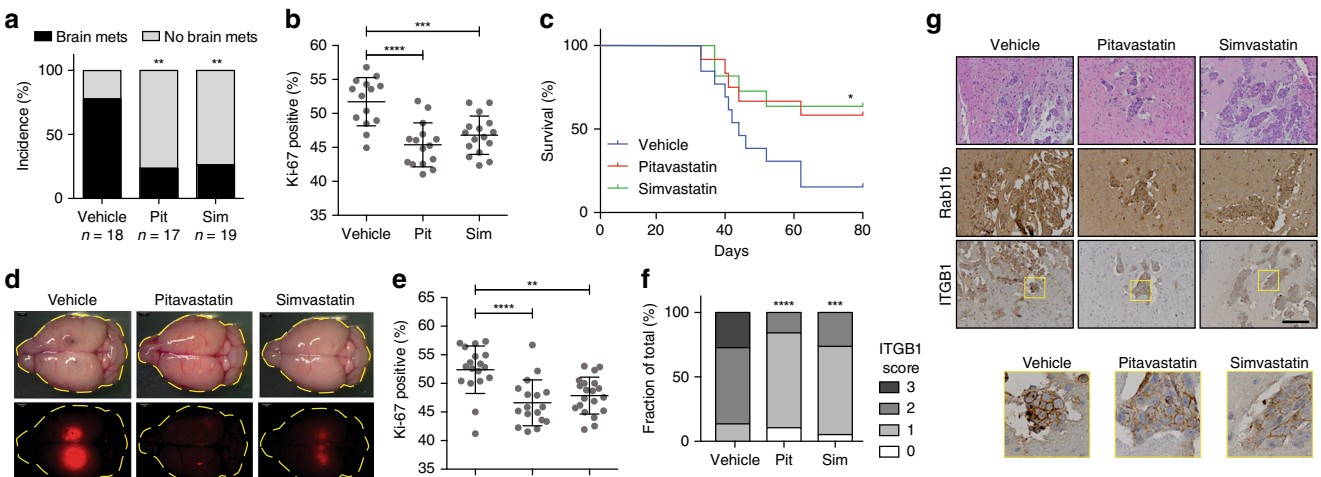

**Fig. 7 Statins decrease breast cancer brain metastasis and improve survival. a–c** MDA-231-Br-GFP cells were intracardially injected, and given daily intraperitoneal injections of vehicle or 1 mg/kg pitavastatin or 5 mg/kg simvastatin. **a** Incidence of brain metastasis determined by visible GFP signal at 28 dpi. Analysis of contingency, Fisher's exact test. **b** Quantification of Ki-67 staining for proliferation. $n = 2$ independent experiments. Bars, mean ± s.d. ANOVA, Tukey's multiple comparison. **c** Survival determined by daily monitoring for the apearance of neurological symptoms or euthanasia criteria. Log rank test. **d–g** MDA-231-tdTomato cells were intracranially injected, and given daily intraperitoneal injections of vehicle or 1 mg/kg pitavastatin or 5 mg/kg simvastatin. **d** Representative images of brain metastasis at 34 dpi. **e** Quantification of Ki-67 staining for proliferation. $n = 2$ independent experiments. Bars, mean ± s.d. ANOVA, Tukey's multiple comparison. **f** Scoring of ITGB1 immunostaining. $n = 2$ independent experiments. Analysis of contingency, Fisher's exact test. **g** Representative images of Rab11b and ITGB1 immunostaining. Scale bar 100 μm. For all panels, *$p < 0.05$, **$p < 0.01$, ***$p < 0.001$, ****$p < 0.0001$.

growth factor receptors such as E-cadherin and EGFR[17,51,52]. We find that Rab11b-mediated recycling of integrin β1 controls surface expression, and therefore ECM ligation, leading to decreased attachment and spreading on integrin β1 ligands. Integrin β1 is the most versatile β isoform[32], with the ability to form heterodimers with a variety of α isoforms. Rab11b-mediated loss of surface integrin β1 leads to decreased activation of adhesion-mediated survival signaling, rendering cells more sensitive to ECM composition. Rab11b knockdown cells are unable to spread and proliferate on decellularized brain matrix, consistent with our finding that loss of Rab11b dramatically reduces brain metastasis formation. Activation of integrin β1 restores attachment and survival of Rab11b cells, suggesting that forced clustering and activation of integrin β1 is able to overcome decreased recycling. Although recycling-mediated localization of specific proteins has been individually studied before, our study demonstrates that, through control of a subset of proteins, Rab11b controls the cell-surface proteome to mediate breast cancer metastatic adaptation to the brain microenvironment. Thus, we propose that the Rab11b-regulated subset of proteins whose expression and localization are dictated by endosomal recycling should be considered the "recycleome", an additional layer of control over cellular behavior. Although we found that Rab11b is the specific isoform up-regulated during BCBM, it is likely that the specific combination of primary cancer type and metastatic microenvironment will dictate the regulation and content of the recycleome. Indeed, the dependence on Rab proteins and effectors has been shown to vary from 2D to 3D culture[53], highlighting the importance of studying trafficking in a specific tumor microenvironmental context.

Breast cancer brain metastases often exhibit a long latent period, and it would be possible to target this critical adaptation period to subsequently prevent metastatic outgrowth[54]. Given the importance of Rab11b in mediating adaptation to the brain metastatic microenvironment, we sought a therapeutic strategy to target Rab11b. The requirement for geranylgeranylation of Rab11b for localization and function provides a unique opportunity to target the mevalonate pathway for non-specific Rab11b inhibition. We demonstrate that statins decrease Rab11b membrane localization, active GTP-bound state, and recycling of both the transferrin receptor and integrin β1. The Rab family of GTPases encompasses over 70 family members, with each member localizing to distinct, but overlapping membranes within the cell[55]. However, all Rab proteins require geranylgeranylation, suggesting that inhibition of the mevalonate pathway and downstream geranylgeranylation with statins could be extended to multiple Rab-mediated clinical scenarios, such as prevention of brain metastasis. Long-term chemoprevention for potential brain metastasis relapse requires the proposed chemopreventive agent to be inexpensive, efficacious, and most importantly, with minimal adverse effects[12]. Well-tolerated HmG-COA reductase inhibiting statins are ideal candidates for direct drug repurposing for brain metastasis prevention. However, given the prevalence of statin use, clinical studies into the effect of statins on cancer have been generally confined to clinical meta-analysis[56,57]. A number of studies have demonstrated the anti-tumor activity of statins via a variety of mechanisms, including inhibition of RhoA, NF-κB, Arf6 or PI3K signaling[44–47]. Statins, particularly lipophilic statins, are believed to be beneficial to patients with breast cancer[58,59], yet the potential for statins in the prevention or treatment of brain metastases has not been explored. In this study, we provide preclinical evidence, based on Rab11b-mediated metastatic adaptation, that repurposing statins could be a practical clinical strategy not only for breast cancer prevention, but also for brain metastasis prevention. Our work identifies Rab11b-mediated recycling as an essential step during brain metastatic adaptation, and outgrowth of DTCs. The preclinical evidence from our study could further guide clinical repurposing of statins by including chemoprevention of breast cancer brain metastasis.

## Methods

**Cell culture**. MDA-MB-231, MDA-MB-468, and BT-20 were purchased from ATCC. MDA-MB-231-Br-EGFP was a generous gift from Patricia Steeg at the National Institute of Health (Bethesda, MD). These lines were maintained in

DMEM/F-12 supplemented with 10% FBS and penicillin-streptomycin. BT549 and HCC38 were purchased from ATCC and maintained in RPMI supplemented with 10% FBS and penicillin-streptomycin. Cancer-associated fibroblast (CAF) cell line was a generous gift from Dr. Zachary Schafer at the University of Notre Dame and was maintained in RPMI supplemented with 10% FBS and penicillin-streptomycin. Primary murine glia were isolated from C57B/6 neonates using aseptic culture technique. Whole brains were passed through a 70 μm filter and spun at 500 G for 5 min to remove myelin. Cells were maintained in high-glucose DMEM supplemented with 10% FBS, 10% horse serum, and penicillin-streptomycin. All cell lines were maintained at 37 °C with 5% $CO_2$. All cells were free of mycoplasma. Human lines were authenticated using STR profiling by Genetica DNA Laboratories.

**Generation of cell lines**. Empty vector and shRab11b (Sigma-Aldrich, SHC001, TRC Numbers: TRCN0000029184, TRCN0000029184) constructs were transfected into HEK293T along with packaging vectors pMD2.G and psPAX2 (Addgene, gift from Didier Trono, 12259 and 12260). Medium with lentiviral particles was collected 48 h after transfection, spun down, 0.45 μm filtered, and added to recipient cell lines. 48 h after addition of virus, cells were selected with puromycin. For siRNA experiments, cells were transfected with 5 or 50 nM siRNA constructs (Sigma-Aldrich, Rab11a: SASI_Hs01_00126206 (#1) and SASI_Hs01_00126207 (#2), Rab11b: SASI_Hs01_00220872 (#1) and SASI_Hs01_00220875 (#2)), and harvested for analysis 72 h post-transfection.

**RNA-sequencing**. Tumors were dissected from brain parenchyma using tdTomato fluorescence as a guide. Total RNA was isolated using the Arcturus PicoPure RNA isolation kit and sequencing libraries constructed using the Ovation RNA-Seq System, both according to manufacturer's directions. Purified libraries were quality assessed with Qubit 2.0 Fluorometer and Agilent 2100 Bioanalyzer analysis. Paired-end transcriptome sequencing ($2 \times 75$ bp) was performed on the Illumina MiSeq sequencer at the Genomics and Bioinformatics Core Facility, University of Notre Dame. Base calling was performed using Real Time Analysis (RTA) v1.17.21.3 (Illumina). Output of RTA was converted into FastQ format with the Bcl2FastQ conversion software v1.8.4 (Illumina). Trimmed reads were aligned to reference human and mouse transcriptomes (Supplementary Fig. 1). Reads that aligned uniquely to the human reference genome were assigned to the tumor gene expression count table. Reads that aligned uniquely to the mouse reference genome were assigned to the microenvironment gene expression count table. All other reads were removed from downstream analysis. Differential expression was conducted using the cufflinks[60] and DESeq2 pipelines[61] as described.

**Fly stocks and genetics**. The fly tumor line expressing oncogenic Ras[V12], dsRNAi targeting *discs large* (Dlg[RNAi]), and GFP under control of the UAS promoter was obtained from M. Willecke[18], and has the full genotype eyflp; UAS-Ras[V12], UAS-Dlg[RNAi(v41134)]/CyO, Gal80[(BL#9491)]; act>CD2 > Gal4, UAS-GFP[S65T] (Supplementary Fig. 2b). To identify RNAi stocks for screening, small, medium and large brain metastases collected at 40 dpi were treated as independent tests of the same hypothesis (tumors had escaped dormancy and were proliferating, regardless of size) and the combined significance was computed on a gene-wise basis using Fisher's combined probability test. Genes with a Fisher's combined *p*-value < 0.05 that were up-regulated in the 40 dpi samples were subjected to *Drosophila* ortholog identification using the *Drosophila* RNAi screening center (DRSC) Integrative Ortholog Prediction Tool (DIOPT). For each human gene the top *Drosophila* ortholog was identified for further analysis. Publicly available RNAi lines were then identified using the DRSC Updated Targets of RNAi Reagents tool (UP-TORR). 448 RNAi lines, representing *Drosophila* orthologs of 108 human genes were obtained from the Bloomington *Drosophila* Stock Center and the Vienna *Drosophila* RNAi Center[62] for analysis.

**Fly tumor screen**. For each RNAi line, 15 female virgins from the tumor genotype were crossed to 8 males from the RNAi line. Flies were left overnight to lay eggs, and progeny were collected on the sixth day after egg laying, when larvae were wandering but not yet pupariating. Larvae were collected in 50% glycerol in water and placed at −20 °C for 15 min prior to imaging. Larvae were arrayed on a plastic dish and imaged on an EVOS FL cell imaging system using the GFP filter cube and transmitted light. A minimum of 15 larvae were imaged for each cross. Transmitted light images were thresholded and used as ROIs for calculation of GFP integrated intensity. SSMD scores were calculated for each line with respect to the negative control (yw), and the positive control (shPTEN). Image and statistical analysis were conducted using custom MATLAB scripts.

**Animal care and use**. All animal use was carried out in accordance with protocols approved by the Notre Dame Institutional Animal Care and Use Committee and were in compliance with the relevant ethical regulations regarding animal research. NOD.Cg-Rag1tm1MomIL2rgtm1Wjl/SzJ (007799/NRG) and C57BL/6J (000664/Black 6) mouse lines were purchased from The Jackson Laboratory and bred in house, with breeders refreshed directly from The Jackson Laboratory annually. For intracranial injections, $2.5 \times 10^4$ cells were injected in 690 nL Hank's buffered saline solution, without calcium, magnesium and phenol red. Bilateral injections were made midway between bregma and lambda, 2 μm off the midline suture. For

intracardiac and intracarotid injections, $2 \times 10^5$ cells were injected in 100 μL Hank's buffered saline solution, without calcium, magnesium and phenol red. For mammary fatpad and tail vein injections, $2.5 \times 10^5$ cells were injected in 50 μL Hank's buffered saline solution, without calcium, magnesium and phenol red. 5-10 animals were used per experimental group, and sample sizes were determined using power analysis, with expected effect size based on prior experience with metastatic animal models. For statin treatment, animals were randomly assigned to vehicle or treatment groups. Beginning two days post injection, animals were given daily intraperitoneal injections of 100 μL vehicle (0.1% hydroxypropyl methylcellulose), pitavastatin (1 mg/kg in vehicle), or simvastatin (5 mg/kg in vehicle).

**Human tissue microarray**. Human tissue microarrays for normal tissue (MBN481), and brain metastases (GL861) were purchased from US Biomax.

**Immunohistochemistry**. Following deparaffinization and rehydration, epitopes were retrieved by boiling slides for 10 min at 100 C in sodium citrate buffer (10 mM sodium citrate, 0.05% Tween-20, pH 6.0). Slides were incubated for 1 h at room temperature in primary antibody (Rab11, Abcam ab3612; Rab11b, Thermo-Fisher PA5-31348; K8, Abcam ab53280; Ki-67, Cell Signaling Technology 9027; ITGB1, Cell Signaling Technology 9699). Primary antibodies were detected using the VECTASTAIN Elite ABC HRP Kit (Vector Laboratories), followed by detection with ImmPACT DAB (Vector Laboratories) following manufacturer's directions. Images were taken with an Olympus BX43.

**Immunocytochemistry**. Cells were plated on glass coverslips and allowed to adhere overnight. Coverslips were fixed in 4% paraformaldehyde, permeabilized with 0.1% Triton X-100, blocked with 1% BSA, and incubated with primary antibody (Rab11, Abcam ab3612; ITGB1, Development Studies Hybridoma Bank AIIB2; 12G10 ITGB1, Abcam ab30394) for 1 h at room temperature. Primary antibodies were detected using appropriate fluorescent secondary antibodies for 1 h at room temperature. Coverslips were stained with phalloidin for F-actin and DAPI for nuclei, and mounted. Images were taken with a Leica DM5500 for coverslips, and a Zeiss LSM 710 inverted confocal microscope for decellularized matrix.

**qPCR**. Total RNA was isolated using RNAzol, and reverse transcribed using Verso cDNA Synthesis Kit. qPCR was performed using 2x SYBR Green qPCR Master Mix on an Eppendorf Mastercycler ep realplex thermocycler with primers: Rab11b F 5′-GTACTACCGTGCAGTGG-3′; Rab11b R 5′-TCCAAGGCTGAGGTCTCGAT-3′; Rab11a F 5′-CTTCGGCCCTAGACTCTACA-3′; Rab11a R 5′-TTCTGACAGCACT GCACCTT-3′; Rab25 F 5′-GAGAAGAGGGCCTGTTGCAT-3′; Rab25 R 5′-CCTA GTCTGTGAGGGGTGGA-3′; Actin F 5′-CCTCGCCTTTGCCGATCC-3′; Actin R 5′-GGCCATCTCTTGCTCGAAGT-3′. Relative expression was determined using the 2CT method with logarithmic transformation.

**Rab11b activation**. A 132 bp fragment of RAB11FIP3 corresponding to amino acids 712-756[28] was amplified from RAB11FIP3 sequence verified cDNA (Dharmacon, MHS6278-202833480) with primers FIP3 F 5′-AGCTCCGTCTCCCGAG AT-3′ and FIP3 R 5′-CTACTTGACCTCCAGGA-3′ and cloned into pGEX-4T-1. RAB11FIP3-GST was expressed in BL21(DE)3 cells, purified using glutathione high-capacity magnetic beads, and stored at 4 °C. For Rab11b activation assay, cell lysates were incubated with RAB11FIP3-GST beads for 1 h at 4 °C, rinsed, boiled, and run on a SDS-polyacrylamide gel.

**Internalization and recycling**. For transferrin receptor internalization and recycling, cells were depleted of serum transferrin (Tf) via serum starvation for 30 min in medium with 0.5% BSA. Medium was replaced with 5 μg/mL Tf-Alexa 647 (Thermo Fisher, T23366) in 0.5% BSA in serum-free medium. For internalization, at each timepoint cells were washed with PBS, surface Tf-Alexa 647 removed via acid stripping buffer (0.5% glacial acetic acid, 500 μM NaCl), fixed in 4% paraformaldehyde and transferred to ice for analysis. For recycling, the Tf trafficking pathway was loaded with Tf-Alexa 647 via 1 h incubation with Tf-Alexa 647 at 37 °C, acid stripped, and medium replaced with full medium containing 50 μg/mL unlabeled transferrin (Sigma-Aldrich, T8158). At each timepoint, cells were fixed and transferred to ice for flow cytometry detection. For co-culture experiments, cancer cells were positively labeled with Alexa Fluor-488-CD44 (BioLegend, 103015). For surface protein biotinylation, surface proteins were biotinylated with Sulfo-NHS-SS-Biotin for 1 h at 4 °C. At each timepoint, cells were permeabilized with 0.1% Triton X-100 for total protein, or left non-permeabilized for surface protein, biotin stained with Biotin-PE (BioLegend, 409003), and subjected to flow cytometry detection. For integrin β1 recycling, surface protein was antibody labeled (ITGB1, Cell Signaling Technology 9699) in HBSS for 30 min on ice. Unbound antibody was washed away, and cells were suspended in prewarmed medium for 1 h at 37 °C. At each timepoint, cells were fixed, transferred to ice, stained with Alexa-488 to detect surface ITGB1, and subjected to flow cytometry detection. Flow cytometry was performed on a Beckman-Coulter FC500, and analysis was performed using the R package FlowCore[63]. Representative gating strategies are available in Supplementary Fig. 10.

**Protein biotinylation and isolation**. Surface proteins were biotinylated using the Pierce Cell Surface Protein Isolation Kit according to the manufacturer's instructions. Briefly, surface proteins were biotinylated with Sulfo-NHS-SS-Biotin, quenched and lysed. For immunoblotting, lysates were incubated with Neutravidin, and isolated biotinylated proteins were quantitated, boiled and run on a SDS-polyacrylamide gel. For Rab11b IP, cell-surface proteins were biotinylated with Sulfo-NHS-SS-Biotin and give 24 h to internalize and recycle proteins. Cells were lysed using a non-denaturing lysis buffer and biotinylated proteins were pulled down with Neutravidin. Protein complexes were boiled and run on a SDS-polyacrylamide gel.

**Immunoblotting**. For general immunoblotting, cells were lysed in RIPA lysis buffer and protein concentrations determined with BCA assay. Normalized lysates were boiled and run on SDS-polyacrylamide gels. To examine signaling following reinitiation of adhesion, cells were suspended for 1 h at 37 °C to downregulate adhesion-mediated signaling, and plated on cell culture plates. To examine signaling in response to pLL and Type I Collagen, cell culture plates were coated with 0.1% pLL or 5 μm/cm$^2$ Type I Collagen for 1 h at room temperature. Cells were suspended for 1 h at 37 °C and plated on coated plates. For integrin inhibition/activation, pLKO.1 and shRab11b cells were incubated for 1 h in suspension at 37 °C with 5 μg/mL P5D2 (Abcam, ab24693) or 12G10 (Abcam, ab30394), respectively, and plated in medium containing P5D2 or 12G10. Separation of soluble and insoluble fractions was done using Triton X-114[64,65]. Briefly, samples were lysed in 2% Triton X-114 and allowed to phase separate. The aqueous (soluble) phase was removed, and the insoluble fraction was washed several times prior to immunoblotting. All lysates were resolved with 8-15% SDS-polyacrylamide gels, and transferred to PVDF (0.2 μm, GE Healthcare) or nitrocellulose (0.45 μm, GE Healthcare) membranes as appropriate. Primary antibodies include: Rab11b (Thermo Fisher, PA5-31348), ITGB1 (Cell Signaling Technology, 9699), FAK (Cell Signaling Technology, 13009), pFAK (Cell Signaling Technology: Tyr397, 8556; Tyr576/577, 3281, Tyr925, 3284), Erk1/2 (Cell Signaling Technology, 4695), pErk1/2 (Cell Signaling Technology, 4370), GFAP (Cell Signaling Technology, 12389), actin (Cell Signaling Technology, 3700). HRP-conjugated secondaries from Cell Signaling or Pierce were used, followed by detection with SuperSignal PLUS Pico or Femto ECL.

**Soft agar**. Six well plates were coated with 1.5 mL of 0.5% Noble agar, and overlaid with 25,000 cells in 0.4% Noble agar in full medium. Cells were treated as required, and medium was changed twice weekly for two or three weeks of growth. For integrin inhibition/activation, cells were incubated for 1 h in suspension at 37 °C with 5 μg/mL P5D2 or 12G10. Cultures were fed with medium containing 5 μg/mL P5D2 or 12G10 for the duration of the experiment. For statin treatment, cells were fed with medium containing 1 or 10 μM pivastatin (MedChem Express, HY-B0144), simvastatin (Sigma-Aldrich, 1612700), or DMSO vehicle control for the duration of the experiment. For metabolite rescue, cells were fed with medium containing 10 μM pivastatin or simvastatin, in addition to 100 μM mevalonolactone (Sigma-Aldrich, M4667) or 10 μM geranylgeranylpyrophosphate (Sigma-Aldrich, G6025) for the duration of the experiment. At endpoint, all wells were fixed with 4% paraformaldehyde, and stained with 0.05% crystal violet solution. 10 images per well were taken with a Leica MDG41 dissecting microscope.

**In-gel protein digestion**. Equal amounts of cell-surface proteins were resolved in SDS-PAGE gels. Each lane was divided into 10 equal sections, and cut into ~1 mm cubes for in-gel digestion. Gels were washed with 50:50 (v/v) acetonitrile (ACN)/ 25 mM ammonium bicarbonate (ABC). After washing, gels were dried, and proteins reduced with 10 mM dithiothreitol (DTT) at 37 °C for 1 h followed by alkylation using 55 mM iodoacetamide (IAA) in the dark at room temperature for 1 h. Samples were enzymatically digested using sequence grade Lys-C/Trypsin in the Barocycler NEP2320 (Pressure Biosciences, Inc.) at 50 °C under 20,000 psi for 1 h. After digestion, peptides were recovered using 60% ACN/5% trifluoroacetic acid (TFA) in purified water with sonication in an ice bath. Recovered peptides were pooled based on protein molecular weight distribution into 3 samples/lane prior to drying in vacuum centrifugation. Recovered peptides were re-suspended in 10 μl of 97% purified water/3% ACN/0.1% formic acid (FA) and 5 μl was used for LC-MS/MS analysis.

**Data analysis of proteomic raw files and bioinformatic analysis**. Analysis was performed using a Dionex UltiMate 3000 RSLC Nano System coupled to a Q Exactive™ HF Hybrid Quadrupole-Orbitrap Mass Spectrometer. Peptides were loaded onto a 300 μm × 5 mm C18 PepMap™ 100 trap column and washed for 5 min with 98% purified water/2% ACN/0.01% FA at a flow rate of 5 μl/min. After washing, the trap column was switched in-line with a 75 μm x 50 cm reverse phase Acclaim™ PepMap™ RSLC C18 analytical column heated to 50°, and peptides were separated using a 120 min linear gradient. The flow rate was 300 nl/minute with a mobile phase A of 0.1% FA in water and a mobile phase B of 0.1 % FA in 80% ACN. The method started at 2% B and reached 10% B in 5 min, 30% B in 80 minutes, 45% B in 93 min, and 100% B in 93 minutes. The column was held at 100% B for the next 5 min before being returned to 2% B where it was equilibrated for 20 min. Samples were injected into the QE HF through the Nanospray Flex™

Ion Source fitted with an emitter tip from New Objective. MS data were collected between 400 and 1600 *m/z* using 120,000 resolutions at 200 m/z, 100 ms maximum injection time, and 15 s dynamic exclusion. The top 20 precursor ions were fragmented by higher energy C-trap dissociation (HCD) at a normalized collision energy of 27%. MS/MS spectra was acquired using the Orbitrap at a resolution of 15,000 at 200 m/z and a maximum injection time of 20 ms. LC-MS/MS data were analyzed using MaxQuant software (v. 1.6.2.10) with the Andromeda search engine[66]. Spectra were searched against the human protein sequence database from UNIPROT retrieved on 05/09/2018 and a common contaminant database for protein identification and relative quantification. A minimal length of six amino acids was required in the database search. The precursor mass tolerance set to 10 ppm, MS/MS fragment ion tolerance was set to 20 ppm, and enzyme specificity for trypsin and LysC allowing up to two missed cleavages. Oxidation of methionine (M) was defined as a variable modification, and carbamidomethylation of cysteine (C) was defined as a fixed modification. The 'unique plus razor peptides' were used for peptide quantitation. The false discovery rate (FDR) of both peptides and proteins identification was set at 0.01. Proteins labeled either as contaminants or reverse hits were removed from the analysis. Identified proteins were quantified using label free MS1 quantitation. Bioinformatic analysis was performed in R (version 3.6.0) using packages available through Bioconductor. ENSEMBL IDs, GO terms, and GO Slim terms were annotated using biomaRt[67]. Proteins with the GO Slim "Plasma Membrane" annotation were considered surface proteins, and were used for further analysis. Pairwise t-tests with post-hoc calculation of the false discovery rate (FDR) using the Benjamini and Yekutieli procedure were conducted in R. GO term incidence was quantified, and transmembrane domains were identified using the prediction algorithm TMHMM[68]. Heatmaps and Cleveland plots were generated using ggplot2[69].

**Adhesion assay**. Forty eight well plates were coated with ECM proteins for 1 h at room temperature as follows: phenol-red free, growth factor-reduced Matrigel (5%; Corning, 356231), Type I Collagen (5 μg/cm$^2$; Santa Cruz, sc-136157), Type IV Collagen (5 μg/cm$^2$; Santa Cruz, sc-29010), Fibronectin (5 μg/cm$^2$; EMD Millipore, 341631), Laminin (5 μg/cm$^2$; Fisher Scientific, CB-40232). Well were rinsed and cells were allowed to adhere. At endpoint, cells were fixed with 10% trichloroacetic acid, and stained with 0.4% sulforhodamine blue (SRB). SRB was solubilized with 10 mM Tris, and read at 554 nm on a BioTek Synergy plate reader.

**Brain matrix decellularization and culture**. Mice were anesthetized with isoflurane, and intracardially perfused with PBS. Brains were removed and cut into 1.5 mm coronal sections (Coronal Brain Matrix, Harvard Apparatus). Brains were decellularized as described[36]. Briefly, brain sections were taken through washes in 4% sodium deoxycholate, DNAse I, 3% Triton X-100, water and PBS all containing penicillin/streptomycin and amphotericin B. Removal of brain parenchymal cells was confirmed using DAPI staining as described. Decellularized brain slices were incubated in medium overnight, and cells were plated and allowed to adhere.

**Image processing and analysis**. All image processing and analysis was done in ImageJ and FIJI[70]. For integrin β1 activation, cell outlines were selected using phalloidin staining as a guide, and the integrated intensity for active integrin β1 (12G10) was determined. Cell area was determined in the same ROI, using the area contained within the phalloidin boundary. Cell length was determined by drawing a straight line along the longest dimension of the cell. To determine the number and size of soft agar colonies, calibrated images were automatically thresholded and the size and number of particles was determined. For EdU and Ki-67 quantification the number of EdU or Ki-67 positive nuclei was divided by the total number of nuclei for each field. Rab11 and ITGB1 IHC staining was qualitatively determined manually.

**Statistical analysis and reproducibility**. Statistical analysis was performed using GraphPad Prism (version 6) and R (version 3.6.0). The following statistical tests were used: Fisher's combined test, strictly standardized mean difference, two-sided t-test, ANOVA with Dunnett's multiple comparison test, two-way ANOVA with Tukey's or Sidak's multiple comparison test, analysis of contingency (Fisher's exact test, two-tailed), analysis of covariance. For column and dot plots, error bars are the average ± standard deviation unless otherwise indicated. For box plots, the box represents the first and third quartiles, the line is the median, the whiskers extend 1.5 times the interquartile range, and all outlying points are plotted individually as dots. All animal experiments were repeated at least twice, with H&E and IHC images representative of staining performed for samples from at least 3 independent animals per experiment. All immunoblots are representative of three independent experiments.

**Reporting summary**. Further information on research design is available in the Nature Research Reporting Summary linked to this article.

## Data availability

The raw RNA-sequencing data are deposited to the National Center for Biotechnology Information Gene Expression Omnibus (GEO) and are available under accession

GSE134405. The raw LC-MS/MS data are deposited to MassIVE and are available under ID MSV000085408 [https://massive.ucsd.edu/ProteoSAFe/dataset.jsp?task=eedd7dd103174b419624256206634cff]. A reporting summary for this article is available as a Supplementary Information file. The source data underlying Figs. 1f–g, 2b–c, e, f, h, 3a, b, d, h, i, 4d, k m, p, 5a–i, k–l, 6b–f, j–l, n–o, 7a-c, e, f, and Supplementary Fig. 4b and 9c are provided as a Source Data file. Source data are provided with this paper.

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

## Acknowledgements

This work was funded by an Advancing Basic Cancer Research grant from the Walther Cancer Foundation (S.Z. and J.J.Z.), DOD W81XWH-15-1-0021 (S.Z.) and NIH grants R01CA194697, R01CA222405 (S.Z.), F32CA210583 (E.N.H.), R03CA212964 (J.L.), R01CA115316 (C.D.S.), TL1TR001107 (E.N.H., awarded by Indiana CTSI, A. Shekhar, PI), an ENSCCII predoctoral fellowship from the Walther Cancer Foundation (P.M.S) and Indiana CTSI Core Pilot Fund (Cohort 16, S.Z. and U.K.A.). We would like to acknowledge and thank the Dee Family endowment (S.Z.). We would also like to thank members of the Zhang and D'Souza-Schorey labs for scientific insight and support. All sample preparation and LC/MS/MS analysis was performed at the Purdue Proteomics Facility. The Q Exactive Orbitrap HF mass spectrometer and the UltiMate 3000 HPLC system used for this study were purchased with generous funding from the Purdue Office of the Executive Vice President for Research and Partnership. We thank the TRiP at Harvard Medical School (NIH/NIGMS R01-GM084947) for providing transgenic RNAi fly stocks used in this study, which were obtained from the Bloomington Drosophila Stock Center (NIH P40OD018537) and the Vienna Drosophila Resource Center (VDRC, [www.vdrc.at]). We are grateful for the use of the following core facilities: Notre Dame Genomics and Bioinformatics Core Facility, Notre Dame Freimann Life Sciences Center, Indiana University School of Medicine South Bend Imaging and Flow Cytometry Core.

## Author contributions

Conceptualization, E.N.H., J.J.Z., S.Z.; Methodology, E.N.H., M.D.B., U.K.A., J.L., S.Z.; Software, E.N.H., M.D.B., P.M.S., A.T.L., J.L.; Formal Analysis, E.N.H., M.D.B., P.M.S., U.K.A., A.T.L.; Investigation, E.N.H., M.D.B., M.E.J., J.W.C., I.H.G., V.H., U.K.A.; Writing Original Draft, E.N.H.; Writing Review & Editing, E.N.H., S.Z.; Supervision, C.D.S., J.J.Z., S.Z.; Funding Acquisition, E.N.H., C.D.S., J.J.Z., S.Z.

## Conflict of Interest

The authors declare no competing interests.
