## [Peer Review File · Nature Communications]

Reviewers' comments:

Reviewer #1 (Remarks to the Author): Expertise in integrin recycling

This is an interesting and novel study that links a member of the Rab11 family, Rab11b, to brain metastasis in breast cancer. The study adopts a broad range of approaches, using *Drosophila*, mouse and human tissue approaches to suggest that Rab11b is upregulated in breast cancer brain metastases, and combines these with aiming and biochemistry to provide evidence that Rab11b controls the surface levels of $\beta 1$ -integrin to promote brain metastasis. This fits broadly with the current understanding of Rab11 function in invasion and metastasis, and in control of integrin function, adding to the literature by demonstrating a previously unknown function for Rab11b in brain metastasis. The study also provides data to suggest that statins could be used as an approach to prevent brain metastasis in breast cancer models. Most of the experiments are well controlled, but the authors have a tendency to overstate their conclusions and I think there are several major issues the authors need to address.

Major points

Figure 1: The data are well presented, but will the complete list of 125 genes identified in mouse models be presented as a supplementary table?

In Figure 1G, mRNA levels of Rab11 are compared in MDA-MB-231 from fat pad tumors versus brain mets. Is this Rab11a and Rab11b? The data for just Rab11b needs to be presented, and data for Rab11a would provide a useful control.

Figure 2: I'm worried about the specificity of antibodies for Rab11b, and without a very good antibody working for western blot, IF and IHC it is very hard for the authors to support the data showing that Rab11b expression is increased at the protein level. In this figure, IHC data are presented purporting to show either 'Rab11' or 'Rab11b'. Rab11a and b show very high sequence identity, and only a few antibodies claim to be selective for Rab11a or b. The authors do not provide information on the source of their antibodies (did they develop their own? Are they commercial?), and there is no evidence to suggest that the antibodies are specific for Rab11b in IHC or immunofluorescence ((Figures 2D, 2H, 3F, 3G, 6O...)). It is essential that the authors provide evidence that the antibodies they use recognize Rab11b in each the applications used.

Figure 2H: I'm a bit confused by the use of Rab11-FIP3 pulldowns. Rab11-FIPs bind to Rab11a and Rab11b (also to Rab25), and the affinities are pretty similar. FIP3 can even bind to Arf6 too. There is nothing specific about this approach for identifying Rab11b and in all the lots presented (including 6H) it looks like the level of Rab11b pulled down with FIP3 is mirrored by the level of protein in the whole cell lysate. I'm not sure this tells us anything other than where there is more protein, there is more active protein.

Figure 3: The authors have used to shRNAs to knockdown Rab11b, but only one does so efficiently ('84'). I don't know if the data using '86' is useful at all- the mRNA levels are not significantly

effected, and the effect on protein level of Rab11b is inconsistent (3A, 3B and 3E). Some data suggest '86' has an effect on the phenotype observed (e.g. Figure 3H) whereas other phenotypes are not influenced by shRNA 86 (Figure 3I, Figure 4....). Is shRNA 84 causing an off target effect? Does shRNA 84 also deplete Rab11a? The authors need another knockdown or knockout approach specific to Rab11b to confirm the major phenotypes in brain metastasis are due to Rab11b depletion.

Fig3 H/I: do these correspond to the intracranial or intracardiac model? It would be more encouraging to see quantitative data from both models.

Figure 4: The data using shRNA84 convincingly show a difference in TFNR recycling upon Rab11b depletion- is the data shown the level of intracellular TFN-647 at each timepoint? This could do with better explanation. What is confusing is that the effect is the same in the parental MDA-231 versus MDA231-Br, which should express more Rab11b? Similarly, co-culture with Glia increases Rab11b levels, but has no effect on TFNR recycling or on levels of activated b1 (4N)? These data seem to imply that the level of Rab11b is not especially important in its function? Are the levels of active b1 or surface b1 integrin different between parental MDA-231 and MDA-231Br?

Figure 5: I can't see how the authors have shown that Rab11b recycling of b1 integrin is necessary for survival here, they should at least perform a b1 integrin recycling experiment to show that recycling is likely to account for the alteration in b1 surface levels. P5D2 and 12G10 both increase pFAK levels in control versus Rab11b knockdown cells?

5C-F: This would benefit from quantification of 3 independent experiments, and a more complete explanation of the data the authors choose to include. Again, the effect of Rab11b shRNA is similar for parental and Br-met cells, which suggests levels of Rab11b do not fully describe its function. pAkt levels seem to be maintained in MDA231-Br upon Rab11b knockdown, whereas pFAK decreases. Is this consistent?

Figure 6: Pitavastatin/simvastatin could work through several different mechanisms, and many other proteins (including the Rab family). The authors need to show the specificity of this pathway- do these drugs have no effect upon knockdown of Rab11b for example? What is the effect of pitavastatin/simvastatin on b1 integrin levels and b1 integrin recycling? Does 12G10 rescue spreading etc upon pitavastatin/simvastatin treatment, as it rescues Rab11b knockdown?

Minor points:

More detailed methods are needed, e.g. with recycling assay, sources of antibodies, isolation of CAFs and glial cells, details of shRNA sequences and Rab11b specific qPCR primers etc...

Reviewer #2 (Remarks to the Author): Expertise in brain metastasis

Authors of manuscript NCOMMS-19-28371 claim Rab11b to be functional mediator of metastatic adaptation by using proteomic analysis. Also reveal lipophilic statins to prevent activity of Rab11b.

This revision comments is focused on specific paragraphs included in three main sections of manuscript:

ABSTRACT

It is stated: "and identify Rab11b as a functional mediator of metastatic adaptation"

It is doubtful according to my comments below.

RESULTS

Here is the manuscript main hypothesis:

"Geranylgeranylpyrophosphate is generated by the mevalonate pathway, which is inhibited by HMG-CoA reductase inhibitors such as statins. Thus, we postulate that statin treatment could suppress Rab11b activation, an essential step for breast cancer adaptation to the brain metastatic microenvironment."

On the contrary, the authors observe:

"Given that statins inhibit membrane localization of Rabs, statin treatment did not alter Rab11b expression as expected".

Such observation does not support their hypothesis of statin suppression the Rab11b activation

DISCUSSION

Their conclusion on manuscript page 10:

"we demonstrate the pre-clinical efficacy of statin treatment for BCBM, and provide a mechanistic rationale for this efficacy, through inhibition of Rab11b geranylgeranylation"

On page 11, the concluding statement:

“However, all Rab proteins require geranylgeranylation, suggesting that inhibition of the mevalonate pathway and its downstream geranylgeranylation with statins could be extended to multiple Rab-mediated clinical scenarios, such as prevention of brain metastasis.”

Is very generous and experimental weakly supported

On page 12, they insist with the statements:

“In this study, we further provided a strong pre-clinical rationale, based on Rab11b-mediated metastatic adaptation mechanisms, that repurposing statins could be a clinically practical strategy for brain metastasis prevention.”

And

“Our study provides a mechanistic rationale for the use of statins in chemoprevention of breast cancer brain metastasis.”

These have a weak support since they do not consider the already reported information on the mechanistic behavior of statins, described in references below that revealed:

- 1) Statins block tumor cell growth in vitro and in vivo by inhibiting production of isoprenoids (dolichol, GPP and FPP).
- 2) These observations have led several investigators to hypothesize that statins might inhibit the growth of a variety of tumor cell types, including prostate, gastric, and pancreatic carcinoma, as well as colon adenocarcinoma, neuroblastoma, glioblastoma, mesothelioma, melanoma, and acute myeloid leukemia cells

Some key references neither mentioned nor discussed in the manuscript:

- 1)Katja Hindler,^a Charles S. Cleeland,^b Edgardo Rivera,^c Charles D. Collarda. The Role of Statins in Cancer Therapy. *The Oncologist* 2006;11:306–315 and references therein.
- 2)Maja Osmak,*Cancer Letters* 324 (2012) 1–12
- 3)Patrizia Gazzero et al, *Pharmacol Rev* 64:102–146, 2012
- 4)Amelia J. McFarland et al, *Int. J. Mol. Sci.* 2014, 15, 20607-20637

Manuel Valiente

Reviewer #3 (Remarks to the Author): Expertise in drosophila (neural) and screens

By combining temporal transcriptional analysis of breast cancer brain metastasis formation in mice at different time points with functional screening of hundreds of genes in *Drosophila*, authors elegantly identify genes that are not only transcriptionally dysregulated, but functionally are responsible to drive tumorigenesis and metastasis. Thus, they show that breast cancer cells significantly up-regulate Rab11b, a small GTPase localized in the endosomal recycling center, in brain metastatic cells compared to the primary site. Mechanistically, the authors convincingly probed that Rab11b-mediated control of the cell surface proteome, including recycling of integrin $\beta 1$ enables successful interaction with the brain extracellular matrix and mechanotransduction-activated survival signaling and proliferation of metastatic cells. Furthermore, they showed that administration of pitavastatin and simvastatin effectively inhibit geranylgeranylation of Rab11b, thereby preventing membrane localization, activation and function, and more importantly decreasing breast cancer brain metastasis. This article is a nice example of how complementary analytical tools and functional studies in different model species can shed the light on the mechanisms operating in breast cancer brain metastases.

The article is very well written and nicely executed; therefore, I strongly suggest to consider this manuscript for publication in *Nature Communications* after some essential revisions.

Essential revisions:

-Although the authors indicate that RasV12, Dlg RNAi tumors develop in the eye disc and progressively invade into adjacent brain tissue in fly larvae (Figure 1D and Figure S2B), a real description of this relevant finding is missing in this study. These relevant results are only shown with a scheme (Figure 1D), and dissections of control, tumorous and rescued tumorous imaginal discs must be shown. Alternatively, representative images of those requested experiments, using the analysis used in Figure 1E with a better resolution that will allow to distinguish GFP+ cells in distant tissues in whole larvae, could also be shown and will strengthen these results, and the use of these model organisms to study metastases *in vivo*.

-Authors in figure 1E show examples of shPTEN and yw with GFP larvae as examples of positive and negative controls, respectively. Authors should clarify if these examples also include RasV12, Dlg RNAi in the genetic background as previously shown as models of tumorous conditions in Willecke et al., 2011. If this not the case, authors should include a picture of a RasV12 Dlg RNAi whole larvae as a positive control in this figure 1E and annotate the correct genotypes of the larvae shown.

Same annotations must be done in figure S2C. Current genotypes are misleading and authors must annotate RasV12, Dlg RNAi in a vertical column on the left to thus, compare tumorous conditions RasV12, Dlg RNAi (first column) with RasV12, Dlg RNAi + Pten RNAi (second column), RasV12, Dlg RNAi+ PSM6 RNAi (third column) and RasV12, Dlg RNAi.+ Rab11 RNAi larvae (fourth column).

Point-by-point response to reviewers' comments

Re: NCOMMS-19-28371

We would like to thank the editor and the reviewers for their valuable and constructive comments. In this revised manuscript we have addressed each of the reviewer's comments, leading to the addition of over 15 new panels of data, more complete descriptions of existing data, and additional description of methods and reagents used. These changes have significantly strengthened and improved the manuscript. Below are our responses to the reviewer's comments, as well as *reviewer comments in size 10 italics*, and changes to the manuscript text in blue.

REVIEWER #1 (Expertise in integrin recycling)

This is an interesting and novel study that links a member of the Rab11 family, Rab11b, to brain metastasis in breast cancer. The study adopts a broad range of approaches, using Drosophila, mouse and human tissue approaches to suggest that Rab11b is upregulated in breast cancer brain metastases, and combines these with aiming and biochemistry to provide evidence that Rab11b controls the surface levels of $\beta 1$ -integrin to promote brain metastasis. This fits broadly with the current understanding of Rab11 function in invasion and metastasis, and in control of integrin function, adding to the literature by demonstrating a previously unknown function for Rab11b in brain metastasis. The study also provides data to suggest that statins could be used as an approach to prevent brain metastasis in breast cancer models. Most of the experiments are well controlled, but the authors have a tendency to overstate their conclusions and I think there are several major issues the authors need to address.

Major points

Figure 1: The data are well presented, but will the complete list of 125 genes identified in mouse models be presented as a supplementary table?

Response: We agree with the reviewer that this information should be provided, and we have included this gene list as Supplemental Table 1.

In Figure 1G, mRNA levels of Rab11 are compared in MDA-MB-231 from fat pad tumors versus brain mets. Is this Rab11a and Rab11b? The data for just Rab11b needs to be presented, and data for Rab11a would provide a useful control.

Response: We agree with the reviewer that differentiating between the three Rab11 family members is important. Figure 1g shows Rab11b mRNA levels, and we have updated Figure 1g to indicate that qPCR was performed for Rab11b. We have updated the manuscript text (page 5, lines 123-127) to make this distinction clear:

“We identified the top 20 *Drosophila* genes based on Aii.SSMD, and examined the expression of their human homologs in samples from MDA-231 human breast cancer xenograft primary tumors and brain metastases. Among the top 3 *Drosophila* genes with the lowest Aii.SSMD (RAB11, SNRPD2, MRPL37), the human homolog of RAB11, RAB11B, was found to be increased 18-fold in brain metastasis tissue (Fig. 1g).”

To study the specificity of Rab11b up-regulation, we also analyzed mRNA expression of Rab family members, Rab11a, Rab11b and Rab25, in cells in culture, primary tumors and brain metastases (Figure 2b). These data show that only Rab11b is increased in the brain microenvironment. In addition, comparing cells in culture to

brain metastases for MDA-231, MDA-231-Br, and MDA-468 cell lines shows elevated Rab11b levels for all cell lines in brain metastases (Figure 2c), with no change in Rab11a or Rab25 levels (Figure S3a).

Figure 2: I'm worried about the specificity of antibodies for Rab11b, and without a very good antibody working for western blot, IF and IHC it is very hard for the authors to support the data showing that Rab11b expression is increased at the protein level. In this figure, IHC data are presented purporting to show either 'Rab11' or 'Rab11b'. Rab11a and b show very high sequence identity, and only a few antibodies claim to be selective for Rab11a or b. The authors do not provide information on the source of their antibodies (did they develop their own? Are they commercial?), and there is no evidence to suggest that the antibodies are specific for Rab11b in IHC or immunofluorescence ((Figures 2D, 2H, 3F, 3G, 6O...)). It is essential that the authors provide evidence that the antibodies they use recognize Rab11b in each the applications used.

Response: We thank the reviewer for this comment - reagents were listed in a supplementary table that was inadvertently omitted from submission. All antibodies used are commercially available. Antibodies and reagents have been added to the methods (**text in red in revised manuscript**). The antibody used to detect Rab11b is specific for Rab11b, as confirmed by the manufacturer. The product details and use in our manuscript are provided here for the reviewer's convenience:

Rab11 - Abcam, ab3612 - IHC (Figure 2a)

Rab11b - Thermo Fisher, PA5-31348 - ICC/IHC/immunoblot (all other figure panels)

Figure 2H: I'm a bit confused by the use of Rab11-FIP3 pulldowns. Rab11-FIPs bind to Rab11a and Rab11b (also to Rab25), and the affinities are pretty similar. FIP3 can even bind to Arf6 too. There is nothing specific about this approach for identifying Rab11b and in all the lots presented (including 6H) it looks like the level of Rab11b pulled down with FIP3 is mirrored by the level of protein in the whole cell lysate. I'm not sure this tells us anything other than where there is more protein, there is more active protein.

Response: The reviewer is correct, Rab11FIP3-GST will bind to many activated (GTP-bound) proteins, including Rab11 family members and Arfs. The specificity of GTP-Rab11b detection in Figure 2h comes from probing the immunoprecipitated lysates for Rab11b using Rab11b specific antibody (Thermo Fisher, PA5-31348), not general Rab11. In this way, the only signal detected on the immunoprecipitated protein complex is the amount of GTP-Rab11b.

The reviewer is also correct that in most contexts within the manuscript we see increased Rab11b levels due to the brain context, which leads to up-regulation of Rab11b (as in Figure 2h). The exception to this is in Figure 6e, where treatment with pitavastatin or simvastatin decreases activated Rab11b, without affecting total Rab11b protein levels, suggesting that increased Rab11b-GTP with increased total Rab11b is not an artifact of the experimental system. To quantitatively examine the relative increase of Rab11b-GTP as compared to total Rab11b, we quantitated blots from two independent experiments for a total of eight independent 7 dpi brain metastasis samples, including the blots shown in Figure 2h. As shown in PTP Figure 1, the quantification confirms that Rab11b (normalized to actin) is increased in brain metastases. Importantly, normalization of Rab11b-GTP levels to normalized Rab11b levels demonstrates that there is a 23% increase in activated Rab11b, suggesting an increased activated proportion of Rab11b. This new quantification data has been added to Supplemental Figure 3, and the manuscript text has been updated to include (page 6, lines 154-158):

“To determine whether the induction of Rab11b expression leads to increased activation of Rab11b, we incubated naïve brain or dissected MDA-231 brain metastasis lysates with Rab11FIP3-GST, which specifically

binds Rab11b-GTP. We found a 76% increase in expression of total Rab11b, confirming our IHC results, as well as a 23% increase in the proportion of active Rab11b-GTP (Fig. 2h, Fig. S3c-d).”

PTP Figure 1. Quantification of Rab11b and Rab11b-GTP induction in brain metastases.

(A) Rab11b divided by actin for each sample. Samples from each blot normalized to MDA-231.

(B) Rab11b-GTP divided by Rab11b/actin ratio shown in (A). Samples from each blot normalized to MDA-231. *Points*, individual samples from two independent experiments. Two-tailed t-test.

Figure 3: The authors have used two shRNAs to knockdown Rab11b, but only one does so efficiently ('84'). I don't know if the data using '86' is useful at all- the mRNA levels are not significantly effected, and the effect on protein level of Rab11b is inconsistent (3A, 3B and 3E). Some data suggest '86' has an effect on the phenotype observed (e.g. Figure 3H) whereas other phenotypes are not influenced by shRNA 86 (Figure 3I, Figure 4....). Is shRNA 84 causing an off target effect? Does shRNA 84 also deplete Rab11a? The authors need another knockdown or knockout approach specific to Rab11b to confirm the major phenotypes in brain metastasis are due to Rab11b depletion.

Response: shRab11b-86 decreases Rab11b levels, albeit not to the extent of shRab11b-84, as the reviewer notes. We included this construct because it supports the specificity of Rab11b knockdown, in part by showing a dose-dependent response, in that shRab11b-86 decreases breast cancer brain metastasis, but not to the extent that shRab11b-84 does. The extent to which shRab11b-86 decreases the mRNA level varies slightly between experimental repeats - ranging from 20-40% knockdown. To more fully present this variation, we have aggregated three separate experiments into Figure 3a and 3d (and PTP Figure 2), and show that shRab11b-86 statistically significantly decreases Rab11b mRNA.

To address the reviewer's question about the effect of shRab11b constructs on other Rab11 family members, we are now including qPCR for all three Rab11 family members. As the reviewer noted above, the Rab11 family members share a high degree of homology, and we used qPCR to differentiate between the isoforms. This data shows that the shRab11b-84 is specific for Rab11b, and does not reduce Rab11a or Rab25 mRNA (PTP Figure 2). Interestingly, we found that the shRab11b-86 construct moderately reduced Rab25 level in the MDA-231 cell line, but not in the MDA-231-Br or MDA-468 cell lines. This data has been added to Figure S4, and the following text has been added to the manuscript (page 6, lines 167-170).

“Given the high degree of homology between Rab11 family members, we sought to determine the specificity of the shRab11b constructs. We show that shRab11b-84 is specific to Rab11b (Fig. S4c), while shRab11b-86 appears to moderately target Rab25 in MDA-231 cells only (Fig. S4c).”

PTP Figure 2. Efficacy and specificity of Rab11b shRNA constructs

(A) Normalized mean RAB11B expression in MDA-231 cells, relative to pLKO.1 empty vector. Three independent experiments. ANOVA with Dunnett's multiple comparison test.

(B) Normalized RAB11B expression in MDA-231 cells cultured alone or with primary murine glia for three days, relative to pLKO.1 alone. Three independent experiments. Two-way ANOVA with Tukey's multiple comparison test.

(C) Rab11 isoform expression in cells expressing indicated constructs. For each isoform, data presented relative to pLKO.1. ANOVA with Dunnett's multiple comparison test.

For all panels, * $p < 0.05$, ** $p < 0.01$, *** $p < 0.001$, **** $p < 0.0001$.

Consistent with this characterization of shRNA construct efficacy and specificity, the majority of our data showed that shRab11b-86 affects the phenotype under investigation in the same way as shRab11b-84 with a diminished effect magnitude, with one exception being the transferrin receptor recycling assay presented in Figure 4a. This experiment investigates the rate at which the internalized transferrin receptor is recycled to the cell surface. For both shRab11b constructs, all internalized transferrin receptor is returned to the cell surface, the difference between knockdown and control cells is a decreased rate at which transferrin receptor is recycled in shRab11b cells. For shRab11b-84 cells, despite an approximate 80% knockdown of Rab11b (Figure 3), the decreased rate of transferrin receptor recycling remained small across all cell lines examined (Figure 4a, Figure S5a). For the less efficient shRab11b-86 construct, which causes only a 20-40% knockdown of Rab11b, any change in transferrin receptor recycling is too subtle to be detected, with the exception of the MDA-468 cell line, which exhibits a statistically significant decrease in transferrin receptor recycling with shRab11b-86 (Figure S5a). Given the small effect size observed with dramatic repression of Rab11b, it is unsurprising that less severe depletion of Rab11b doesn't change the rate of transferrin receptor recycling, particularly given the importance and robustness of transferrin receptor recycling. Due to the lack of effect with shRab11b-86, as well as the potentially confounding effect of shRab11b-86 on Rab25 in MDA-231 cells, we have elected to remove this data from Figure 4a, and focus only on the effect induced with shRab11b-84.

Fig3 H/I: do these correspond to the intracranial or intracardiac model? It would be more encouraging to see quantitative data from both models.

Response: The data in Figure 3h and i corresponded to MDA-231-Br cells intracardially injected. We agree with the reviewer that quantitative data from both models would be useful for readers. We have therefore quantitated Ki-67 staining for the MDA-231 intracranial injection model (PTP Figure 3, left). In response to the

reviewer's previous question about the efficacy of shRab11b-86, we quantitated additional animals for the MDA-231-Br intracardiac injection model (PTP Figure 3, right). When additional samples were added, we found that the trend for shRab11b-86 to decrease proliferation became significant. Both of these panels have been added to Figure 3.

PTP Figure 3. Ki-67 quantification in intracranial and intracardiac brain metastasis models.

Quantitation of Ki-67 staining in MDA-231 (intracardiac) and MDA-231-Br (intracranial) brain metastasis models. ANOVA with Dunnett's multiple comparison test. For all panels, * p < 0.05, *** p < 0.001, **** p < 0.0001.

Figure 4: The data using shRNA84 convincingly show a difference in TFNR recycling upon Rab11b depletion- is the data shown the level of intracellular TFN-647 at each timepoint? This could do with better explanation. What is confusing is that the effect is the same in the parental MDA-231 versus MDA231-Br, which should express more Rab11b? Similarly, co-culture with Glia increases Rab11b levels, but has no effect on TFNR recycling or on levels of activated b1 (4N)? These data seem to imply that the level of Rab11b is not especially important in its function? Are the levels of active b1 or surface b1 integrin different between parental MDA-231 and MDA-231Br?

Response: The reviewer is correct, the data in Figure 4a and b shows the level of intracellular transferrin receptor/transferrin-Alexa 647. Briefly, cells were exposed to transferrin-Alexa 647 and allowed to saturate the pathway for 60 minutes. Surface fluorescence was then stripped with acid stripping buffer, leaving only internal transferrin-Alexa 647. At each of the time points, the surface was again acid stripped, and the amount of transferrin-Alexa 647 remaining inside the cells was quantified. To make this more clear, we have updated plots to specify that the y axis is "FL4 - Intracellular Transferrin-647", and updated the manuscript text to include the following (page 6, lines 182-183):

"We examined the rate at which TfR was recycled by assaying the amount of fluorescently labeled transferrin retained in cells following 1 hr loading with fluorescent transferrin."

And expanded the description in the methods (page 18, lines 546-553):

"For transferrin receptor internalization and recycling, cells were depleted of serum transferrin (Tf) via serum starvation for 30 min in medium with 0.5% BSA. Medium was replaced with 5 µg/mL Tf-Alexa 647 (Thermo Fisher, T23366) in 0.5% BSA in serum-free medium. For internalization, at each time point cells were washed with PBS, surface Tf-Alexa 647 removed via acid stripping buffer (0.5% glacial acetic acid, 500 µM NaCl), fixed in 4% paraformaldehyde and transferred to ice for analysis. For recycling, the TfR trafficking pathway was loaded with Tf-Alexa 647 via 1 hr incubation with Tf-Alexa 647 at 37C, acid stripped, and medium replaced with full medium containing 50 µg/mL unlabeled transferrin (Sigma-Aldrich, T8158). At each time point, cells were fixed and transferred to ice for flow cytometry detection."

The reviewer is also correct that MDA-231-Br express slightly more Rab11b than MDA-231 cells (Figure 2), but knockdown of Rab11b with shRab11b-84 leads to similar decreases in the amount of Rab11b. Given the importance of TfR recycling in cellular homeostasis, as well as the expression of Rab11a in all cell lines examined, it isn't surprising that loss of Rab11b merely decreases the rate at which TfR is recycled, without loss of TfR recycling. In addition, there is also a "fast" recycling pathway, which doesn't involve the Rab11 isoforms, which could explain the moderate effect of shRab11b-84 on TfR recycling. We have expanded our description of these results to make this more clear (pages 5-6, lines 183-187):

"Compared with control cells, loss of Rab11b led to slower TfR recycling (Fig. 4a, Fig. S5a), although TfR was fully recycled in all cell lines examined, consistent with the requirement for TfR recycling in cellular homeostasis. Internalization of TfR was not affected (Fig. S5b), consistent with the role of Rab11b in regulating transport of proteins from the endosomal recycling center (ERC) to the cell surface. These results confirm that shRab11b functionally perturbs endosomal recycling."

The reviewer is exactly right, and we agree that the data presented shows that a glia-mediated increase in Rab11b does not increase TfR recycling. Although Rab11b is involved in TfR recycling, brain-mediated up-regulation of Rab11b, while dramatic in terms of Rab11b levels, does not dramatically change TfR recycling. This result formed the basis of our hypothesis that there could be brain microenvironment specific recycling cargo (recycleome) mediated by Rab11b. We suggest that the level of Rab11b is especially important for the recycling of specific cargo proteins, and use proteomics to identify proteins specifically recycled by Rab11b, including integrin β_1 . We have reworked this section of the results to clarify this (page 6, 187-191):

"Interestingly, there is not a detectable increase of TfR recycling when cells are cultured with either primary glia or CAF cells (Fig. 4b, Fig. S5c), despite the strong glial-mediated induction of Rab11b (Fig. 2e). This suggests that brain-mediated up-regulation of Rab11b might regulate a specific subset of cell surface proteins, rather than globally altering recycling of all cargo proteins."

Figure 5: I can't see how the authors have shown that Rab11b recycling of β_1 integrin is necessary for survival here, they should at least perform a β_1 integrin recycling experiment to show that recycling is likely to account for the alteration in β_1 surface levels. P5D2 and 12G10 both increase pFAK levels in control versus Rab11b knockdown cells?

Response: We agree with the reviewer that it is important to examine Rab11b-mediated recycling of integrin β_1 . We have now examined the rate of integrin β_1 recycling in control and shRab11b cells. We have added the following text to describe the new data we have added to revised Figure 4 i-m (main text pages 7-8, lines 218-222):

"To determine if loss of Rab11b alters integrin β_1 recycling, surface integrin β_1 was antibody labeled and cells were suspended for 1 hr to downregulate adhesion signaling and allow internalization of integrin β_1 . Analysis of labeled integrin β_1 recycled to the surface reveals that control cells rapidly recycle integrin β_1 , while shRab11b cells exhibit lower surface levels (Fig. 4l) and the rate of integrin β_1 recycling is significantly decreased (Fig. 4m)."

PTP Figure 4. Integrin β 1 recycling.

(A) Surface integrin β 1 recycling in control or shRab11bMDA-231 cells. Two-way ANOVA with Sidak's multiple comparison test.

(B) Linear regression of data in (P). Analysis of covariance.

For all panels, * $p < 0.05$, ** $p < 0.01$, *** $p < 0.001$, **** $p < 0.0001$.

As for the question regarding P5D2 and 12G10 antibodies, we answer as follows: Yes, the inhibitory P5D2, and the activating 12G10 antibodies both lead to increased FAK phosphorylation. P5D2 inhibits integrin β 1 activity by blocking ligand binding; however, as the reviewer notes, a known 'side effect' of P5D2 is phosphorylation of FAK, please see references below. There is a loss of integrin attachment-mediated signaling due to the functional loss of ligand binding, but there is phosphorylation of FAK. To the best of our knowledge, the reason for this is not known, but despite FAK phosphorylation, P5D2 is considered to reliably and robustly inhibit integrin β 1-mediated attachment, and downstream signaling.

Yokosaki Y, Palmer EL, Prieto AL, Crossin KL, Bourdon MA, Pytela R, Sheppard D. The integrin alpha 9 beta 1 mediates cell attachment to a non-RGD site in the third fibronectin type III repeat of tenascin. *J Biol Chem.* 1994 Oct 28;269(43):26691-6. PubMed PMID: 7523411.

Xia H, Nho RS, Kahm J, Kleidon J, Henke CA. Focal adhesion kinase is upstream of phosphatidylinositol 3-kinase/Akt in regulating fibroblast survival in response to contraction of type I collagen matrices via a beta 1 integrin viability signaling pathway. *J Biol Chem.* 2004 Jul 30;279(31):33024-34. Epub 2004 May 27. PubMed PMID: 15166238.

5C-F: This would benefit from quantification of 3 independent experiments, and a more complete explanation of the data the authors choose to include. Again, the effect of Rab11b shRNA is similar for parental and Br-met cells, which suggests levels of Rab11b do not fully describe its function. pAkt levels seem to be maintained in MDA231-Br upon Rab11b knockdown, whereas pFAK decreases. Is this consistent?

Response: We agree with the reviewer that this data was not well explained. To improve the data presentation we have quantified three independent experiments in Figure 5c, and removed Akt and Erk blots from Figure 5c to better focus on FAK activation, which is the intended focus. In this figure, we sought to determine if signaling downstream of integrin engagement is disrupted with shRab11b without regard for cell intrinsic differences in downstream signaling pathways. There is not a dramatic difference in Rab11b levels between MDA-231 and MDA-231-Br cell lines, regardless of shRab11b knockdown. However, as the reviewer noted, pFAK decreases consistently across cell lines with shRab11b. We have streamlined the data and results to make this panel focus on Rab11b's effect on pFAK and revised manuscript text accordingly (page 8, lines 241-247):

"To examine signaling downstream of integrin-mediated attachment, we suspended cells for 1 hr to force down-regulation of adhesion-mediated signaling, and found that shRab11b induced a severe defect in post-attachment activation of focal adhesion kinase (FAK) (Fig. 5c). Control cells exhibit sustained

phosphorylation at residues 397 and 925 for up to 6 hrs (Fig. 5c, Fig. 7Sb), demonstrating auto-phosphorylation and activation of FAK. shRab11b cells exhibit decreased phosphorylation at all of the sites, including residue 925, which is involved in cell migration and protrusions, suggesting that shRab11b cells fail to spread (Fig. 5b) due to decreased focal adhesion formation.”

PTP Figure 5. Quantification of pFAK signaling during adhesion.

Cells were suspended for 1 hr at 37C, then plated to allow adhesion complex formation. Representative immunoblots shown in Figure 5C. Quantification of pFAK in MDA-231 (top) and MDA-231-Br (bottom) samples. Data presented relative to FAK normalized to actin for three independent experiments. Two-way ANOVA with Sidak’s multiple comparison test. For all panels, * p < 0.05, ** p < 0.01, *** p < 0.001, **** p < 0.0001.

It is interesting that MDA-231-Br cells maintain pAkt with Rab11b knockdown and decreased pFAK. It is possible that the Br cell line derivative has developed an alternative or compensatory pathway to drive sustained pAkt. Understanding the complexity of Akt signaling adaptation will require future studies. Given the confusing cell line-specific differences in Akt activation downstream of integrin β 1, we have removed Akt blots from Figure 5e and Figure 5f as well. Instead, we have focused on activation of FAK, and activation of Erk, which are relatively consistent across all cell lines examined. We have updated the manuscript text to expand on these results (pages 8-9, lines 253-259):

“We next examined the effect of shRab11b on FAK and Erk signaling during adhesion to an integrin β 1 ligand. When control and shRab11b cells are plated on pLL, although both lines have low activation, shRab11b cells exhibit decreased activation of FAK and Erk (Fig. 5e), consistent with their decreased ability to spread. Across multiple cell lines, control cells dramatically increase activation of FAK and Erk when plated on Col I, while shRab11b cells exhibit decreased phosphorylation of FAK and Erk, suggesting that loss of surface integrin β 1 induces a defect in ligation to Col I leading to decreased adhesion-mediated signaling (Fig. 5e).”

Figure 6: Pitavastatin/simvastatin could work through several different mechanisms, and many other proteins (including the Rab family). The authors need to show the specificity of this pathway- do these drugs have no effect upon knockdown of Rab11b for example? What is the effect of pitavastatin/simvastatin on β 1 integrin levels and β 1 integrin recycling? Does 12G10 rescue spreading etc upon pitavastatin/simvastatin treatment, as it rescues Rab11b knockdown?

Response: We agree with the reviewer that given the importance of the mevalonate pathway, inhibition with statins exerts pleiotropic effects on cell behavior. For this reason, we endeavored to make clear in our manuscript not that statins specifically inhibit Rab11b, but that Rab11b inhibition is one of the downstream effects of statins. To make this more clear, we have rearranged the data in Figure 6, and performed the experiments suggested by the reviewer. We reference several mechanisms by which statins have been shown to inhibit tumor growth, and these references are now closer to the beginning of this section. We examined the effect of statins on control and shRab11b cells in soft agar as suggested by the reviewer and found that statin treatment is largely ineffective in Rab11b knockdown cells (PTP Figure 6). This data has been added to Figure 6d and Figure S8b, and the manuscript text has been updated (page 10, lines 303-306):

“We next sought to determine if there was a synergistic effect between statin treatment and Rab11b knockdown. In soft agar assays, we found a small but insignificant decrease in colony size, with no decrease in colony number (Fig. 6d, Fig. S8b).”

PTP Figure 6. Statin treatment does not further inhibit shRab11b-mediated reduced growth in soft agar.

Quantification of number and size of colonies for cells grown in soft agar for three weeks with vehicle or statins. Ten fields per condition. Two-way ANOVA with Sidak’s multiple comparison test. For all panels, * p < 0.05, ** p < 0.01, *** p < 0.001, **** p < 0.0001.

As reviewer recommended, we also examined the effect of pitavastatin and simvastatin on integrin β 1 recycling, and found that the rate of integrin β 1 recycling to the cell surface is decreased with statin treatment (PTP Figure 7a). We also activated integrin β 1 with 12G10 in combination with statin treatment to determine if activation of integrin β 1 signaling rescues statin treatment. We found that 12G10 restores cell spreading and protrusion formation (PTP Figure 7b), as well as proliferation (PTP Figure 7d) of MDA-231 cells plated on the decellularized brain matrix. These results have been added to Figure 6, and the manuscript text updated (page 10, lines 315-325):

“We next sought to determine if statins would prevent integrin β 1 recycling and adhesion-mediated signaling. Both pitavastatin and simvastatin decreased the rate at which internalized integrin β 1 is recycled to the cell surface (Fig. 6h). When cells are plated on decellularized brain matrix, statin treatment prevents cell spreading and protrusion formation (Fig. 6i-j), decreases activation of integrin β 1 (Fig. 6k), and inhibits proliferation of breast cancer cells on the brain ECM (Fig. 6i), suggesting a role for integrin β 1 downstream of mevalonate pathway inhibition. To confirm the role of integrin β 1, we treated cells with statins alone or in conjunction with an integrin β 1 activating antibody (12G10). We found that activation of integrin β 1 restores cell spreading and protrusion formation on decellularized brain matrix (Fig. 6m-n), as well as proliferation (Fig. 6o). Taken together, this data suggests that both pitavastatin and simvastatin successfully inhibit Rab11b, leading to decreased recycling and activation of integrin β 1.”

PTP Figure 7. Statins decrease integrin $\beta 1$ recycling, and integrin $\beta 1$ activation overcomes statin-mediated growth inhibition

(A) Transferrin receptor recycling in MDA-231 cells treated with vehicle or 1 μM pitavastatin or simvastatin. Two-way ANOVA with Sidak's multiple comparison test.

(B-D) MDA-231 cells treated with 1 μM pitavastatin or simvastatin and 12G10, and allowed to adhere to decellularized murine brain matrix for 48 hrs. **(B)** Representative images of actin (phalloidin, red) protrusions (arrowheads). Scale bar 10 μm . **(C)** Quantification of cell area and longest dimension. ANOVA with Tukey's multiple comparison test. **(D)** Quantification of EdU in ten fields per condition. ANOVA with Tukey's multiple comparison test. For all panels, * $p < 0.05$, ** $p < 0.01$, *** $p < 0.001$, **** $p < 0.0001$.

Minor points:

More detailed methods are needed, e.g. with recycling assay, sources of antibodies, isolation of CAFs and glial cells, details of shRNA sequences and Rab11b specific qPCR primers etc...

Response: We have updated the methods to include these details.

Reviewer #2 (Remarks to the Author): Expertise in brain metastasis

Authors of manuscript NCOMMS-19-28371 claim Rab11b to be functional mediator of metastatic adaptation by using proteomic analysis. Also reveal lipophilic statins to prevent activity of Rab11b.

This revision comments is focused on specific paragraphs included in three main sections of manuscript:

ABSTRACT

It is stated: "and identify Rab11b as a functional mediator of metastatic adaptation"
It is doubtful according to my comments below.

Response: The reviewer's comments are focused primarily on the last element - statin treatment - of the manuscript, where we investigate the preclinical efficacy of statins for the treatment of breast cancer brain metastasis through inhibition of Rab11b geranylgeranylation. Our conclusion that Rab11b is a functional mediator of breast cancer brain metastasis was supported by loss-of-function genetic evidence presented in Figure 3, where we showed that loss of Rab11b decreases the incidence and severity of breast cancer brain metastasis in two models, an intracranial and an intracardiac injection model (Figure 3f-i). The ability of statins to inhibit Rab11b geranylgeranylation, or breast cancer brain metastasis, is not germane to the conclusion that Rab11b is a functional mediator of breast cancer brain metastatic adaptation.

RESULTS

Here is the manuscript main hypothesis:

"Geranylgeranylpyrophosphate is generated by the mevalonate pathway, which is inhibited by HMG-CoA reductase inhibitors such as statins. Thus, we postulate that statin treatment could suppress Rab11b activation, an essential step for breast cancer adaptation to the brain metastatic microenvironment."

On the contrary, the authors observe:

"Given that statins inhibit membrane localization of Rabs, statin treatment did not alter Rab11b expression as expected".

Such observation does not support their hypothesis of statin suppression the Rab11b activation

Response: We respectfully disagree with the reviewer on this point. The amount of active Rab11b is not strictly dependent on the total amount of Rab11b, much as the level of a phosphoprotein is not strictly dependent on the total amount of protein. Geranylgeranylpyrophosphate (GGPP) is generated by the mevalonate pathway, and as such it is known to be decreased when the mevalonate pathway is inhibited with statins. GGPP is required for proteins to be geranylgeranylated, a lipid modification that allows interaction with lipid membranes, similar to myristoylation or palmitoylation. The Rho family of GTPases, including Rab11b, requires geranylgeranylation to interact with endosomal membranes and thus to function in endosomal trafficking^{1,2}. We sought to determine if statins would inhibit Rab11b functional activity through inhibition of GGPP synthesis in the specific context of breast cancer brain metastases.

As the reviewer points out, in Figure 6f we show that neither pitavastatin nor simvastatin treatment changes the total level of Rab11b protein. However, as we show in Figure 6e, both pitavastatin and simvastatin move Rab11b from the insoluble (geranylgeranylated, membrane-bound) to the soluble (non-geranylgeranylated, non-membrane bound) cellular fractions in a dose-dependent manner, demonstrating that statin treatment inhibits the ability of Rab11b to interact with membranes. Consistent with the loss of membrane localization, in Figure 6f we show that there is decreased active Rab11b with both pitavastatin and simvastatin treatment. Thus, we have demonstrated directly that there is less active Rab11b with both pitavastatin and simvastatin treatment (Figure 6f, Rab11FIP3-GST pulldown for Rab11b-GTP). We have rewritten this section of the results to clarify this (page 10, lines 309-312):

"To determine if statins decrease Rab11b geranylgeranylation, we examined the membrane localization of Rab11b. Both pitavastatin and simvastatin moved Rab11b from the insoluble, membrane-bound fraction to the soluble, non-membrane bound fraction in a dose-dependent manner (Figure 6e, Figure S8c), suggesting a loss of Rab11b membrane localization."

DISCUSSION

Their conclusion on manuscript page 10:

“we demonstrate the pre-clinical efficacy of statin treatment for BCBM, and provide a mechanistic rationale for this efficacy, through inhibition of Rab11b geranylgeranylation”

On page 11, the concluding statement:

“However, all Rab proteins require geranylgeranylation, suggesting that inhibition of the mevalonate pathway and its downstream geranylgeranylation with statins could be extended to multiple Rab-mediated clinical scenarios, such as prevention of brain metastasis.”

Is very generous and experimental weakly supported

Response: The ability of statins to inhibit the production of isoprenoids, including geranylgeranylpyrophosphate (GGPP) has been previously reported in the literature³⁻⁶. We are not the first to suggest that statins could inhibit geranylgeranylation of Rabs, or farnesylation of Ras family proteins. The vast majority of reviews, including the four mentioned by the reviewer below, suggest this very concept. Given the established role of statins in suppressing geranylgeranylation of Rab proteins for membrane localization and function, it is reasonable to postulate that statins could be used in multiple Rab-mediated clinical scenarios as part of our discussion.

Specific to the brain metastasis context demonstrated in our study, we show that Rab11b is dramatically up-regulated during breast cancer brain metastasis (Figure 2), and demonstrate that loss of Rab11b decreases breast cancer brain metastasis (Figure 3). Based on our discovery of the importance of Rab11b in breast cancer brain metastasis, we hypothesized that statins could inhibit Rab11b localization and function through inhibition of GGPP synthesis. We went on to show that statins inhibit the membrane localization (Figure 6e), and activation (Figure 6f) of Rab11b. The ability of statins to inhibit breast cancer growth in soft agar (Figure 6b), was rescued by the addition of GGPP (Figure 6c), specifically demonstrating that geranylgeranylation of Rab proteins is important for breast cancer cell growth in soft agar. We also demonstrate that statin treatment decreases breast cancer formation (Figure 7). Taken together, we have shown that pitavastatin and simvastatin inhibit localization and function of Rab11b, which we demonstrated earlier in the manuscript to be important for breast cancer brain metastasis. Based on the multifaceted evidence in our study, therefore, we believe that our statement is a logical extension of both previous literature on statins and Rabs, and the mechanistic work in this manuscript.

On page 12, they insist with the statements:

“In this study, we further provided a strong pre-clinical rationale, based on Rab11b-mediated metastatic adaptation mechanisms, that repurposing statins could be a clinically practical strategy for brain metastasis prevention.”

And

“Our study provides a mechanistic rationale for the use of statins in chemoprevention of breast cancer brain metastasis.”

These have a weak support since they do not consider the already reported information on the mechanistic behavior of statins, described in references below that revealed:

- 1) Statins block tumor cell growth in vitro and in vivo by inhibiting production of isoprenoids (dolichol, GPP and FPP).*
- 2) These observations have led several investigators to hypothesize that statins might inhibit the growth of a variety of tumor cell types, including prostate, gastric, and pancreatic carcinoma, as well as colon adenocarcinoma, neuroblastoma, glioblastoma, mesothelioma, melanoma, and acute myeloid leukemia cells*

Some key references neither mentioned nor discussed in the manuscript:

1)Katja Hindler,a Charles S. Cleeland,b Edgardo Rivera,c Charles D. Collarda. *The Role of Statins in Cancer Therapy. The Oncologist* 2006;11:306–315 and references therein.

2)Maja Osmak,*Cancer Letters* 324 (2012) 1–12

3)Patrizia Gazzero et al, *Pharmacol Rev* 64:102–146, 2012

4)Amelia J. McFarland et al, *Int. J. Mol. Sci.* 2014, 15, 20607-20637

Response: We are fully aware of the large and growing body of work with respect to statins in the treatment of cancer. In our previous submission, we cited multiple primary research articles showing some of the mechanisms by which statins have been demonstrated to inhibit tumor growth (references 60, 61, 62, and 63. Despite the reported anti-tumor activity of statins, to the best of our knowledge, there is no literature that has shown a role for brain metastatic environment-induced Rab11b overexpression in breast cancer brain metastasis, or provided mechanistic rationale to support the use of statins in the treatment or prevention of breast cancer brain metastasis. Our study provided additional mechanistic insights into the previously observed anti-tumor effects of statins by identifying a role for prevention of Rab11b geranylgeranylation in breast cancer brain metastasis. The novelty of our work, particularly the statin-related aspect of the study, centers on the following important findings:

- 1) The anti-tumor efficacy of statins could derive partially from inhibition of Rab11b geranylgeranylation as evidenced by geranylgeranyl pyrophosphate rescue experiments (Figure 6c and Supplementary Fig. 8a) and statin-mediated inhibition of Rab11b activation (Figure 6f). As brain colonized tumor cell dramatically up-regulated Rab11b and such up-regulation is functionally essential for the brain metastasis success. Statin treatment targeting Rab11b is well suited for the treatment of Rab11b-mediated brain metastasis.
- 2) Statins are effective in suppressing brain metastasis (Figure 7a, d), and improving survival (Figure 7c) in our preclinical models of breast cancer brain metastasis. Given the relatively low clinical toxicity of statins, it is reasonable to expect that statins could be used as a potential agent for brain metastasis prevention.

In light of the reviewer's comments, to clarify the novelty of our study further, we have revised our conclusive sentences and highlighted the Rab11b-specific brain metastasis context presented in our study (pages 12-13, lines 394-403).

"A number of studies have demonstrated the anti-tumor activity of statins via a variety of mechanisms, including inhibition of RhoA, NF- κ B, Arf6 or PI3K signaling^{60–63,82–84}. Statins, particularly lipophilic statins, are believed to be beneficial to patients with breast cancer^{78–81}, yet the potential for statins in the prevention or treatment of brain metastases has not been explored. In this study, we provide preclinical evidence, based on Rab11b-mediated metastatic adaptation, that repurposing statins could be a practical clinical strategy not only for breast cancer prevention, but also for brain metastasis prevention. Our work identifies Rab11b-mediated recycling as an essential step during brain metastatic adaptation, and outgrowth of disseminated tumor cells. The preclinical evidence from our study could further guide clinical repurposing of statins by including chemoprevention of breast cancer brain metastasis."

We thank the reviewer for pointing out a number of prior studies on statins in the literature. A comprehensive discussion of the statin literature is more appropriate for an all-inclusive review article, and beyond the scope of our primary research article with a narrower focus. Thus, in this revision, after carefully analyzing the literature regarding statins, we selectively cited the most recent studies/reviews.

We do not find the following literature mentioned by the reviewer to be closely relevant to our study:

1. Hinder et al summarize the preclinical and clinical data on statins in the prevention and treatment of cancer up to 2006, the publication date of their review. They conclude that statins may be beneficial for

cancer patients, stating “Now is the time for appropriately designed clinical trials to underpin the promising data of pre-existing studies.” They reference the work of Farina et al showing that lovastatin inhibits breast cancer lung metastases (Farina HG, Bublik DR, Alonso DF, Gomez DE. Lovastatin alters cytoskeleton organization and inhibits experimental metastasis of mammary carcinoma cells. Clin Exp Metastasis. 2002;19(6):551-9. PubMed PMID: 12405293), but neither reference nor mention any data supporting the use of statins in the treatment or prevention of breast cancer brain metastases.

2. The review by Osmak focuses on clinical trials of statins in the treatment of cancer, and in vitro research combining statins with chemotherapeutics to kill cancer cells of various origins. This review also references the work of Farina et al, but makes no mention of statins in the treatment or prevention of breast cancer brain metastases.
3. The review by McFarland et al focuses on the effect of statins on neurodegenerative diseases, with a limited discussion of primary central nervous system cancers. Although the discussion of statins in neurodegenerative diseases is well-considered, there is a distinct lack of relevance to cancer or metastasis.

We have added citations of the following literature to revised discussion section:

1. The review by Gazzero et al is extensive, covering the effects of statins on both normal tissue and cancer. We thank the reviewer for bringing this review to our attention and agree that it is a good reference for the broad body of literature that exists with respect to statins. We have added a reference to this review in our revised manuscript (ref 84, page 12, line 396)
2. There have been several reviews on the role of statins in cancer, and we cited two which we consider to be more comprehensive and more recent than the above-mentioned reviews (Hinder et al, Osmak et al, and McFarland et al):
 - a. Ref 77 - Gronich, N. & Rennert, G. Beyond aspirin—cancer prevention with statins, metformin and bisphosphonates. Nat Rev Clin Oncol 10, 625–642 (2013).
 - b. Ref 81 - Beckwitt, C. H., Brufsky, A., Oltvai, Z. N. & Wells, A. Statin drugs to reduce breast cancer recurrence and mortality. Breast Cancer Res. 20, 144 (2018).

REVIEWER #3 (Expertise in drosophila and brain)

By combining temporal transcriptional analysis of breast cancer brain metastasis formation in mice at different time points with functional screening of hundreds of genes in Drosophila, authors elegantly identify genes that are not only transcriptionally dysregulated, but functionally are responsible to drive tumorigenesis and metastasis. Thus, they show that breast cancer cells significantly up-regulate Rab11b, a small GTPase localized in the endosomal recycling center, in brain metastatic cells compared to the primary site. Mechanistically, the authors convincingly probed that Rab11b-mediated control of the cell surface proteome, including recycling of integrin β 1 enables successful interaction with the brain extracellular matrix and mechanotransduction-activated survival signaling and proliferation of metastatic cells. Furthermore, they showed that administration of pitavastatin and simvastatin effectively inhibit geranylgeranylation of Rab11b, thereby preventing membrane localization, activation and function, and more importantly decreasing breast cancer brain metastasis. This article is a nice example of how complementary analytical tools and functional studies in different model species can shed the light on the mechanisms operating in breast cancer brain metastases.

The article is very well written and nicely executed; therefore, I strongly suggest to consider this manuscript for publication in Nature Communications after some essential revisions.

Essential revisions:

-Although the authors indicate that RasV12, Dlg RNAi tumors develop in the eye disc and progressively invade into adjacent brain tissue in fly larvae (Figure 1D and Figure S2B), a real description of this relevant finding is missing in this

study. These relevant results are only shown with a scheme (Figure 1D), and dissections of control, tumorous and rescued tumorous imaginal discs must be shown. Alternatively, representative images of those requested experiments, using the analysis used in Figure 1E with a better resolution that will allow to distinguish GFP+ cells in distant tissues in whole larvae, could also be shown and will strengthen these results, and the use of these model organisms to study metastases in vivo.

Response: We appreciate that the reviewer finds our work interesting. The ability of Ras^{V12}/shDlg eye disc tumors to invade into the adjacent central nervous system is not an original finding from this study. The Ras^{V12}/shDlg model was developed by Pagliarini and Xu (Pagliarini RA, Xu T. A genetic screen in Drosophila for metastatic behavior. *Science*. 2003 Nov 14;302(5648):1227-31. Epub 2003 Oct 9. PubMed PMID: 14551319). In the Pagliarini and Xu study, they provided a detailed characterization of invasion and metastasis behavior of this model. In this revised manuscript, we have updated the manuscript text to clarify that Figure 1d and Figure S2b represent an experimental schematic and the full genotype of the flies used, and make clear that invasion into the brain tissue is a known characteristic of the model (page 4, lines 105-110):

“This model overexpresses oncogenic RasV12, an RNAi construct targeting the polarity gene discs large (Dlg), and green fluorescent protein (GFP) in the epithelial Drosophila imaginal eye disc. In this model, tumors develop in the eye disc and progressively invade into adjacent brain tissue³². We identified Drosophila orthologs for the 108 BrainMets Sig.Genes³³, and obtained 448 RNAi fly lines (Figure S2a). A simple genetic cross drives expression of the RNAi construct specifically in the tumorous eye tissue (Figure 1d, full genotype in Figure S2b).”

We understand the reviewer’s interest in metastases outside of the CNS, and we did observe GFP+ cells outside of the CNS in a minority of larvae. We did not pursue either identification or quantification of these metastases for two reasons:

- 1) Robust metastasis to the brain was one reason we chose the model, as our principal interest was to determine which of the genes identified in the RNA-sequencing were functionally driving brain metastasis.
- 2) Metastases outside of the CNS occurred at low frequency and incidence, and therefore analysis of these metastases would have decreased the power of the fly model by significantly increasing the time and resources required for screening experiments.

Authors in figure 1E show examples of shPTEN and yw with GFP larvae as examples of positive and negative controls, respectively. Authors should clarify if these examples also include RasV12, Dlg RNAi in the genetic background as previously shown as models of tumorous conditions in Willecke et al., 2011. If this not the case, authors should include a picture of a RasV12 Dlg RNAi whole larvae as a positive control in this figure 1E and annotate the correct genotypes of the larvae shown.

Same annotations must be done in figure S2C. Current genotypes are misleading and authors must annotate RasV12, Dlg RNAi in a vertical column on the left to thus, compare tumorous conditions RasV12, Dlg RNAi (first column) with RasV12, Dlg RNAi + Pten RNAi (second column), RasV12, Dlg RNAi+ PSM6 RNAi (third column) and RasV12, Dlg RNAi.+ Rab11 RNAi larvae (fourth column).

Response: We apologize for the confusion. All experiments performed in flies were done using crosses between RNAi lines and the Ras^{V12}/shDlg tumor line. The cross to yw was the negative control, showing tumorigenesis in the background Ras^{V12}/shDlg tumor line. We have updated the schematic in Figure 1d to make clear that all crosses are RNAi x Ras^{V12}, Dlg^{RNAi} tumors. We have also updated the SSMD plot in Figure

1e to show the full genetic background of each line. We'd like to thank the reviewer for the suggestions on annotation. Likewise, we have updated Supplemental Figure 2c to clearly indicate that all experimental RNAi lines were crossed into the Ras^{V12}, Dlg^{RNAi} tumor background.

References

1. Joberty, G., Tavitian, A. & Zahraoui, A. Isoprenylation of Rab proteins possessing a C-terminal CaaX motif. *FEBS Lett.* **330**, 323–328 (1993).
2. Resh, M. D. Covalent lipid modifications of proteins. *Curr. Biol.* **23**, R431–435 (2013).
3. Yanae, M. *et al.* Statin-induced apoptosis via the suppression of ERK1/2 and Akt activation by inhibition of the geranylgeranyl-pyrophosphate biosynthesis in glioblastoma. *Journal of Experimental & Clinical Cancer Research* **30**, 74 (2011).
4. Bulhak, A., Roy, J., Hedin, U., Sjöquist, P.-O. & Pernow, J. Cardioprotective effect of rosuvastatin in vivo is dependent on inhibition of geranylgeranyl pyrophosphate and altered RhoA membrane translocation. *American Journal of Physiology-Heart and Circulatory Physiology* **292**, H3158–H3163 (2007).
5. Parikh, A. *et al.* Statin-induced autophagy by inhibition of geranylgeranyl biosynthesis in prostate cancer PC3 cells. *The Prostate* **70**, 971–981 (2010).
6. Hashimoto, A. *et al.* P53- and mevalonate pathway-driven malignancies require Arf6 for metastasis and drug resistance. *J. Cell Biol.* **213**, 81–95 (2016).

Reviewers' comments:

Reviewer #1 (Remarks to the Author):

The authors have clarified most of their methods and reagents, and included new data to show that statins oppose recycling and prevent anchorage independent growth in control, but not Rab11b knockdown cells. This helps to support the main conclusions of the paper.

The following points have not been answered fully:

1) Concerns over Rab11b antibody specificity.

The antibody PA5-31348 (ThermoFisher) was raised against recombinant human Rab11b aa's 1-213. As seen in the alignment below, there is a high degree of similarity between Rab11a and Rab11b at the protein sequence level.

Rab11a MGTRDDEYDYLFKVVLLIGDSGVGKSNLLSRFTRNEFNLESKSTIGVEFATRSIQVDGKTI 60

Rab11b MGTRDDEYDYLFKVVLLIGDSGVGKSNLLSRFTRNEFNLESKSTIGVEFATRSIQVDGKTI 60

Rab11a KAQIWDTAGQERYRAITSAYYRGAVGALLVYDIAKHLTYENVERWLKELRDHADSNIVIM 120

Rab11b KAQIWDTAGQERYRRITSAYYRGAVGALLVYDIAKHLTYENVERWLKELRDHADSNIVIM 120

Rab11a LVGNKSDLRHLRAVPTDEARAFKNGLSFIETSALDSTNVEAAFQILTIEYRIVSQKQ 180

Rab11b LVGNKSDLRHLRAVPTDEARAFKNNLSFIETSALDSTNVEEAFKNILTEYRIVSQKQ 180

Rab11a MSDRRENDMSPSNVPIHVPPTTEN--KPKVQCCQNI 216

Rab11b IADRAAHDESPGNNVVDISVPPTTDGQKPNKLQCCQNL 218

..** .* **.***** * ****.* *.*****:

It looks like the antibody signal decreases upon knockdown of Rab11b when detected by western blot (Figure 3b). ICC and IHC are very different methods, and it is therefore necessary to show that the antibody recognises Rab11b (and not Rab11a) when considering non-denatured protein too. I would be more convinced if the authors show loss of signal by knockdown of Rab11b by ICC. I think this is a relatively straightforward experiment that the authors should include (e.g. using MDA-MB-

231 pLKO.1 vs shRab11b 86). It's harder to show this specificity by IHC, but ICC would help convince that the antibody discriminated between Rab11b and Rab11a when detecting intact, folded protein.

2) Potential off target effects of shRNA

The differences in effect of shRNAs could be explained by an off target effect of shRNA 84. It is good to see that this knockdown does not influence Rab11a levels, but to rule out off target effects they should re-express knockdown resistant Rab11b, or use an independent and equally effective knockdown/knockout and confirm the findings of one or two of the more straightforward key experiments (e.g. integrin/active integrin levels in Figure 4k, n, o; effects on cell area Figure 5b and FAK activity (Figure 5c)).

Reviewer #3 (Remarks to the Author):

The authors have made the appropriate changes in response to my comments and clearly have improved the manuscript. I have no further questions.

Reviewer #4 (Replacement for reviewer#2, Remarks to the Author):

In this revised submission of the manuscript entitled Rab11b-mediated integrin recycling promotes brain metastatic adaptation and outgrowth, the authors present a series of in vivo and in vitro studies demonstrating a novel role for Rab11b in promoting brain metastatic colonization. Additional to extensive addition of new data supporting the hypothesis of Rab1b mediating IntegrinB1 recycling and promoting brain colonization, the authors have provided adequate explanations to the initial comments of reviewer #2, including further clarification in the manuscript.

I consider this revised version is adequate for publication.

Point-by-point response to reviewers' comments

Re: NCOMMS-19-28371

We would like to thank the editor and the reviewers for their valuable and constructive comments. Reviewers 3 and 4 state that our revised manuscript addressed all of their concerns. Reviewer 1 requested additional clarifications about the reagents used. Below are our responses to the reviewer's comments, as well as *reviewer comments in size 10 italics*.

Reviewer#1: (Remarks to the Author)

The authors have clarified most of their methods and reagents, and included new data to show that statins oppose recycling and prevent anchorage independent growth in control, but not Rab11b knockdown cells. This helps to support the main conclusions of the paper.

The following points have not been answered fully:

1) Concerns over Rab11b antibody specificity.

The antibody PA5-31348 (ThermoFisher) was raised against recombinant human Rab11b aa's 1-213. As seen in the alignment below, there is a high degree of similarity between Rab11a and Rab11b at the protein sequence level.

```
Rab11a MGTRDDEYDYLFKVVLIGDSGVGKSNLLSRFTRNEFNLESKSTIGVEFATR SIQVDGKTI 60
Rab11b MGTRDDEYDYLFKVVLIGDSGVGKSNLLSRFTRNEFNLESKSTIGVEFATR SIQVDGKTI 60
*****

Rab11a KAQIWDTAGQERYRAITSAYYRGAVGALLVYDIAKHLTYENVERWLKELRDHADSNI VIM 120
Rab11b KAQIWDTAGQERYRRITSAYYRGAVGALLVYDIAKHLTYENVERWLKELRDHADSNI VIM 120
*****

Rab11a LVGNKSDLRHLRAVPTDEARAFAEKNGLSFIETSALDSTNVEAAFQ TILTEIYRIVSQKQ 180
Rab11b LVGNKSDLRHLRAVPTDEARAFAEKNNLSFIETSALDSTNVEEAFKNILTEIYRIVSQKQ 180
*****

Rab11a MSDRRENDMSPSNNVVPIHVPPTTEN--KPKVQCCQNI 216
Rab11b IADRAAHDESPGNVVDISVPPTTDGQKPNKLQCCQNL 218
:.* * * * * . * * * * * . * * * * * .
```

It looks like the antibody signal decreases upon knockdown of Rab11b when detected by western blot (Figure 3b). ICC and IHC are very different methods, and it is therefore necessary to show that the antibody recognises Rab11b (and not Rab11a) when considering non-denatured protein too. I would be more convinced if the authors show loss of signal by knockdown of Rab11b by ICC. I think this is a relatively straightforward experiment that the authors should include (e.g. using MDA-MB-231 pLKO.1 vs shRab11b 86). It's harder to show this specificity by IHC, but ICC would help convince that the antibody discriminated between Rab11b and Rab11a when detecting intact, folded protein.

Response: We agree with the reviewer, there is a high degree of amino acid similarity between Rab11a and Rab11b. However, a small difference in amino acid sequence could significantly

change the 3D protein folding, charges, and ultimate structure. Many antibodies were produced to recognize the small differences in protein residues, such as phosphorylation and sumoylation.

We would like to point out that the nucleotide sequences are quite divergent, as shown below. , we do wish to highlight that the sequence targeted by shRab11b-84 (TRC clone TRCN0000029184 - highlighted in yellow below) targets a sequence found only in Rab11b. As the reviewer has noted, the decrease of antibody signal upon knockdown of Rab11b using shRab11b-84 when detected by western blot (Figure 3b) suggested that the antibody we used is most likely to be specific to Rab11b.

Rab11a vs Rab11b Similarity : 192/657 (29.22 %)

		M G T R D D E Y D Y L		F K V V L I G	
Rab11a	1	atgggcacccgacgacgagtagcactacct-----ctttaagttg-tccttattg	52		
Rab11b	1	atggggacccgggacgacgagtagcactacctattcaaag--tgggtgctcatcggggact	58		
		M G T R D D E Y D Y L F K V		V L I G D S	
		D S G V G K S N L L S R F T R N E F N L			
Rab11a	53	gagattctggtgttgaaagagtaatctcctgtctcgattactcgaaatgagtttaatc	112		
Rab11b	59	caggcgtgggcaagagcaacctgctgtcgcgcttcacccgcaacgagttcaacctggaga	118		
		G V G K S N L L S R F T R N E F N L E S			
		E S K S T I G V E F A T R S I Q V D G K			
Rab11a	113	tggaaagcaagagcaccattggagtagagtttgaacaagaagcatccaggttgatggaa	172		
Rab11b	119	gcaagagcaccatcggcgtggagttcggcaccgcagcatccaggtggacggcaagacca	178		
		K S T I G V E F A T R S I Q V D G K T I			
		T I K A Q I W D T A G Q E R Y R A I T S			
Rab11a	173	aaacaataaaggcacagatattgggacacagcagggaagagcgatattcgagctataacat	232		
Rab11b	179	tcaaggcgcagatctgggacaccgctggccaggagcgctaccgcgcatcacctccgcgt	238		
		K A Q I W D T A G Q E R Y R A I T S A Y			
		A Y Y R G A V G A L L V Y D I A K H L T			
Rab11a	233	cagcatattatcgtggagctgtaggtgccttattggtttatgacattgctaacaatctca	292		
Rab11b	239	actaccgtggtgcagtgggcgccctgctggtgtacgacatcgccaagcacctgacctatg	298		
		Y R G A V G A L L V Y D I A K H L T Y E			
		Y E N V E R W L K E L R D H A D S N I V			
Rab11a	293	catatgaaaatgtagagcgtggctgaaagaactgagagatcatgctgatagtaacattg	352		
Rab11b	299	agaacgtggagcgtggctgaaggagctgcccggaccacgcagacagcaacatcgctcatca	358		
		N V E R W L K E L R D H A D S N I V I M			

		I M L V G N K S D L R H L R A V P T D E	
Rab11a	353	ttatcatgcttgtgggcaataagagtgatctacgtcatctcagggcagttcctacagatg	412
Rab11b	359	tgctggtgggcaacaagagtgacctgcgccacctgcgggctgtgccactgacgaggccc	418
		L V G N K S D L R H L R A V P T D E A R	
		A R A F A E K N G L S F I E T S A L D S	
Rab11a	413	aagcaagagcttttgcagaaaagaatggtttgcattcattgaaacttcggccttagact	472
Rab11b	419	ggccttggattccacta	478
		A F A E K N N L S F I E T S A L D S T N	
		T N V E A A F Q T I L T E I Y R I V S Q	
Rab11a	473	ctacaaatgtagaagctgcttttcagacaattttaacagagatttaccgcattgtttctc	532
Rab11b	479	acgtagaggaagcattcaagaacatcctcacagagatctaccgatcgtgtcacagaaac	538
		V E E A F K N I L T E I Y R I V S Q K Q	
		K Q M S D R R E N D M S P S N N V V P I	
Rab11a	533	agaagcaaatgtcagacagacgcgaaaatgacatgtctccaagcaacaatgtggttccta	592
Rab11b	539	agatcgcagaccgcgctgccacgacgagtcctccggggaacaacgtggtggacatcagcg	598
		I A D R A A H D E S P G N N V V D I S V	
		H V P P T T E N K P K V Q C C Q N I *	
Rab11a	593	ttcatgttccaccaaccactgaaaacaagccaaaggtgcagtgctgtcagaacatctaa	651
Rab11b	599	tgccgccaccacggacggacagaagcccaacaagctgcagtgctgccagaacctgtga	657
		P P T T D G Q K P N K L Q C C Q N L *	

Furthermore, we have performed the experiment suggested by the reviewer, and examined Rab11b protein levels in control and both shRab11b cell lines using anti-Rab11b antibody (Catalog #PA5-31348, ThermoFisher). As shown in PTP Figure 1 (ICC experiment as reviewer requested), there is a decrease in Rab11b levels in both shRab11b-84 and shRab11b-86 cell lines. Consistent with data included in our manuscript (Figure 3 a-b, Figure S4 a-c), we see that shRab11b-84 significantly decreases Rab11b levels, while shRab11b-86 has a more moderate effect. This is consistent with our qRT-PCR results examining the levels of Rab11a, Rab11b and Rab25 (Figure S4c), and as noted above, although Rab11a and Rab11b share a high degree of protein homology, the RNA is quite divergent, making them easily distinguishable. We have added this data to Figure S4b. Given the reviewer's request below for an additional method of decreasing Rab11b, we have also included an immunoblot showing that two additional siRab11b constructs decrease the level of Rab11b, while two siRab11a constructs show no such decrease. In conclusion, after applying different shRNA/siRNA constructs to differentiate a/b isoform by both WB and ICC methods, we believe the Rab11b antibody (Thermo Fisher, PA5-31348) we used is specific to the "b" isoform of Rab11.

PTP Figure 1. Specificity of Rab11b antibody (ThermoFisher, PA5-31348)

(A) Immunostaining for Rab11b (red) and nuclei (DAPI, blue) in MDA-231 cell lines as indicated.
 (B) Corrected total cellular Rab11b fluorescence (CTCF) determined for individual cells. ANOVA with Dunnett's multiple comparison test. ** $p < 0.01$, **** $p < 0.0001$.
 (C) Immunoblot showing Rab11b expression 72 hr after siRNA transfection.

2) Potential off target effects of shRNA

The differences in effect of shRNAs could be explained by an off target effect of shRNA 84. It is good to see that this knockdown does not influence Rab11a levels, but to rule out off target effects they should re-express knockdown resistant Rab11b, or use an independent and equally effective knockdown/knockout and confirm the findings of one or two of the more straightforward key experiments (e.g. integrin/active integrin levels in Figure 4k, n, o; effects on cell area Figure 5b and FAK activity (Figure 5c)).

Response: We agree that off-target effects of shRNA constructs are a concern, which is why we had initially examined five shRab11b constructs, narrowing down to the two most effective constructs in the manuscript (shRab11b-84 and shRab11b-86). Unfortunately, as shown above, shRab11b-84 targets a region within the coding sequence, and thus we are unable to express a rescue construct as the reviewer suggests. We have obtained two siRNAs targeting Rab11b (Sigma-Aldrich, SASI_Hs01_00220872 - siRab11b #1 and SASI_Hs01_00220875 - siRab11b #2) and performed the experiments suggested by the reviewer. As shown above, both siRNAs effectively reduce the level of Rab11b (PTP Figure 1c), with construct #1 being slightly more effective. We found that siRNA-mediated decrease of Rab11b leads to no change in integrin $\beta 1$ levels (PTP Figure 2a), but a significant decrease in activated integrin $\beta 1$ (PTP Figure 2b-c), consistent with what we found using shRab11b constructs (Figure 4n-p). We also found that both siRab11b constructs significantly decreased the ability of cells to spread and elongate on Collagen I (PTP Figure 3), consistent with our finding using shRab11b constructs (Figure 5b). Taken together, this data suggests that the phenotypes we observed using shRab11b constructs are in fact due to the loss of Rab11b, and not due to off-target effects. These data have been included as a new Supplementary Figure 5.

PTP Figure 2. Loss of active Integrin β1 with siRab11b

(A) MDA-231 cells stained for integrin β1 and actin (phalloidin, to delineate cell boundaries).

(B) MDA-231 cells stained for active integrin β1 and actin (phalloidin, to delineate cell boundaries).

(C) Corrected total cellular active integrin β1 fluorescence (CTCF) determined for individual cells. ANOVA with Dunnett's multiple comparison test. *** $p < 0.001$, **** $p < 0.0001$.

PTP Figure 3. Loss of cell spreading with siRab11b

(A) MDA-231 cells plated on poly L-lysine or Collagen I and stained for actin (phalloidin, red) and nuclei (DAPI, blue).

(B) Quantification of cell length. Line, mean, bars, s.d. ANOVA, Tukey's multiple comparison test. *** $p < 0.001$, **** $p < 0.0001$.

REVIEWERS' COMMENTS:

Reviewer #1 (Remarks to the Author):

The authors have included new data and addressed my comments, including new data. I believe this is a valuable addition to the field.